# Room temperature photochemical synthesis of metal–organic frameworks for enhanced photocatalysis

Yong Wang [1,2,8], Jingzhuo Guan[3,8], Kush Kumar[4], Wanting He[1], Jesus Valdez[5], Ruiqi Yang[1], Guoping Hu [2], Shengyun Huang[2], Audrey Moores [5,6,7], Santosh Kumar Meena [4], Yongfeng Zhou [3], Yannan Liu [3] ✉ & Dongling Ma [1] ✉

The function of metal–organic frameworks (MOFs) is fundamentally governed by their synthesis precision. Here, we report a light-driven strategy enabling ambient-temperature MOFs synthesis (15 °C, 4 hours) for cobalt-porphyrin frameworks (phoPPF-3), overcoming traditional thermal constraints. This approach achieves multidimensional control, manifested in two-dimensional hourglass morphologies and selective $Co^{2+}$-carboxylate coordination that preserves free-base porphyrin cores unattainable conventionally. Resulting phoPPF-3 exhibits enhanced thermal stability and higher photocatalytic activity in benzyl alcohol oxidation and $H_2$ evolution comparing to solvothermal analogues. The methodology demonstrates a certain generality through successful extension to other MOFs. This work marks the demonstration of using photons to initiate and guide MOFs synthesis and establishes a sustainable approach for atomically precise MOFs engineering via photochemical control.

Metal–organic frameworks (MOFs), consisting of metallic ions and organic linkers, have attracted increasing attention due to their tailorable chemistry, tunable porosity, and ordered crystalline structures[1–4]. These structural features make MOFs excellent candidates for photocatalysis[5,6], including water splitting[7–10], pollutant degradation[11], $CO_2$ reduction[12–16], bacteria disinfection[17], and nitrogen fixation[18]. However, the quality of MOFs, including their crystal structure, particle shape, degree of crystallinity, defect levels, and active site availability, is critically dependent on their synthesis pathways. Hydrothermal/solvothermal synthesis, the most common synthetic method of MOFs, typically requires high heat (80–220 °C) and long reaction time (6–72 h), which often cause problems such as redox-

active metal centers oxidization (e.g., $Fe^{2+}$ to $Fe^{3+}$) and performance-limiting defects formation[19–22]. While various alternative synthetic strategies have been explored to accelerate MOFs synthesis with the purpose of gaining higher levels control over structure and morphology, including mechanochemical synthesis[23], microwave-assisted synthesis[24], magnetic-induced heating[25], photothermal-assisted synthesis[26], and ultrasonic method[27], the full performance potential of the resulting MOFs remains unrealized due to irreversible thermal involved metal-node oxidation or inadequate morphology/defect control.

Light-controlled synthesis offers a potential solution. It utilizes light as an energy input to initiate or accelerate chemical reactions

[1]Énergie Matériaux Télécommunications Research Centre, Institut National de la Recherche Scientifique, Varennes, QC, Canada. [2]Ganjiang Innovation Academy, Chinese Academy of Sciences, Ganzhou, China. [3]State Key Laboratory of Synergistic Chem-Bio Synthesis, School of Chemistry and Chemical Engineering, Shanghai Jiao Tong University, Shanghai, China. [4]Department of Chemical Engineering, Indian Institute of Technology Ropar, Rupnagar, India. [5]Facility for Electron Microscopy Research (FEMR), McGill University, Montreal, QC, Canada. [6]Centre in Green Chemistry and Catalysis, Department of Chemistry, McGill University, Montreal, QC, Canada. [7]Department of Materials Engineering, McGill University, Montreal, QC, Canada. [8]These authors contributed equally: Yong Wang, Jingzhuo Guan. ✉e-mail: lyannan@sjtu.edu.cn; dongling.ma@inrs.ca

through mechanisms such as photoinduced electron transfer, photo-mediated atom transfer, and photoinduced energy transfer[28], offering advantages including room temperature reaction, sustainability, and precise spatiotemporal control. Crucially, unlike conventional hydro-thermal/solvothermal methods governed by thermodynamic path-ways, this approach enables a shift towards kinetically controlled processes[29], providing enhanced synthetic control by allowing, for example, instantaneous synthesis-on-and-off switching. Consequently, light-controlled synthesis demonstrates significant potential for pre-paring diverse materials, including organic polymers[30–32], covalent organic frameworks (COFs)[33,34], and inorganic nanoparticles[35–38]. For example, Tanabe et al. reported a living photopolymerization based on photoexcited metal-containing ferrocenophane monomers[32]. This method enables bond-specific monomer activation, enhancing effi-ciency and allowing controlled polymerization with much milder anionic initiators. Choi et al. developed an ultraviolet (UV) light-driven photochemical synthesis of COF-5 at mild conditions, achieving a higher yield and shorter reaction time than conventional solvothermal method[33]. Kim et al. reported a photochemical synthesis of Au nanorods in a micellar solution, demonstrating excellent control over particle size, shape, and uniformity[35]. However, to our knowledge, the synthesis of MOFs via light control remains largely unexplored. This gap likely stems from the inherent complexity and difficulty of applying photochemical processes to MOF synthesis containing both metal ions and organic ligands. Therefore, developing green and controllable light-driven methods for MOF synthesis holds significant scientific importance and is needed to unlock the full potential of MOFs.

Herein, we report the photon-initiated, room-temperature pho-tochemical synthesis of MOFs that directs coordination and mor-phology without external photothermal agents, a key distinction from light-induced heating approaches. Specifically, we demonstrate a visible light-driven synthesis of a cobalt porphyrin framework (phoPPF-3) using tetrakis(4-carboxyphenyl)porphyrin (TCPP), 4,4′-bipyridine (BPY), and cobalt ions, without additional photothermal agents (Fig. 1). Combined experimental and theoretical analyses reveal that TCPP acts as a photoactive ligand, driving metal coordination,

oriented growth, and loose structural assembly under visible light. Compared to solvothermal synthesis (80 °C, 24 h), our method achieves comparable yield (72.8%) in just 8 h at room temperature of 15 °C and reaches high yield of 97.6% at 12 h. While maintaining similar crystallinity to plate-like PPF-3 through traditional solvothermal method, the formed phoPPF-3 shows a three-dimensional (3D) hour-glass morphology. Interestingly, light input induces selective $Co^{2+}$-carboxylate coordination, preserving the free-base porphyrin, unlike solvothermal conditions where $Co^{2+}$ binds both carboxylates and porphyrin N. This results in enhanced thermal stability of phoPPF-3 than PPF-3, verified by laser-integrated transmission electron micro-scopy (TEM) that allows in situ laser heating and real-time observation of structure with a high spatial resolution, and thermogravimetry-differential thermal analysis in conjunction with mass spectrometry (TG/DTA-MS). As a result, phoPPF-3 shows higher photocatalytic activity in the oxidation of benzyl alcohol to benzaldehyde and $H_2$ evolution, potentially benefited from proton-coupled electron transfer (PCET) of free-base porphyrin structure in phoPPF-3. The generality of this synthetic strategy is demonstrated through successful synthesis of other MOFs such as HKUST-1, ZIF-67, and ZIF-8. This work establishes a green, light-driven approach enabling precise control over MOF structure and functionality, opening an avenue for advanced MOF design and promoting their practical applications.

## Results and discussion

### Visible light-driven synthesis and characterization of phoPPF-3

The visible light-driven synthesis of phoPPF-3 was achieved by mod-ifying a conventional solvothermal method used for two-dimensional (2D) PPF-3 nanosheets[39]. As illustrated in Fig. 2a, cobalt nitric ($Co(NO_3)_2$) and TCPP serve as the cobalt source and photoactive organic ligand, respectively. A precursor solution containing $Co(NO_3)_2$, TCPP, polyvinylpyrrolidone (PVP), and BPY in a dimethyl-formamide/ethanol mixture was prepared (Supplementary Fig. 1a). Under typical solvothermal condition (80 °C, 24 h), PPF-3 nanosheets were formed with a yield of 71.3% (Supplementary Fig. 1b). In contrast, irradiating the same precursor solution at 15 °C for 4 h using a 100 W light-emitting diode (LED) lamp ($\lambda = 420$ nm) resulted in hourglass-

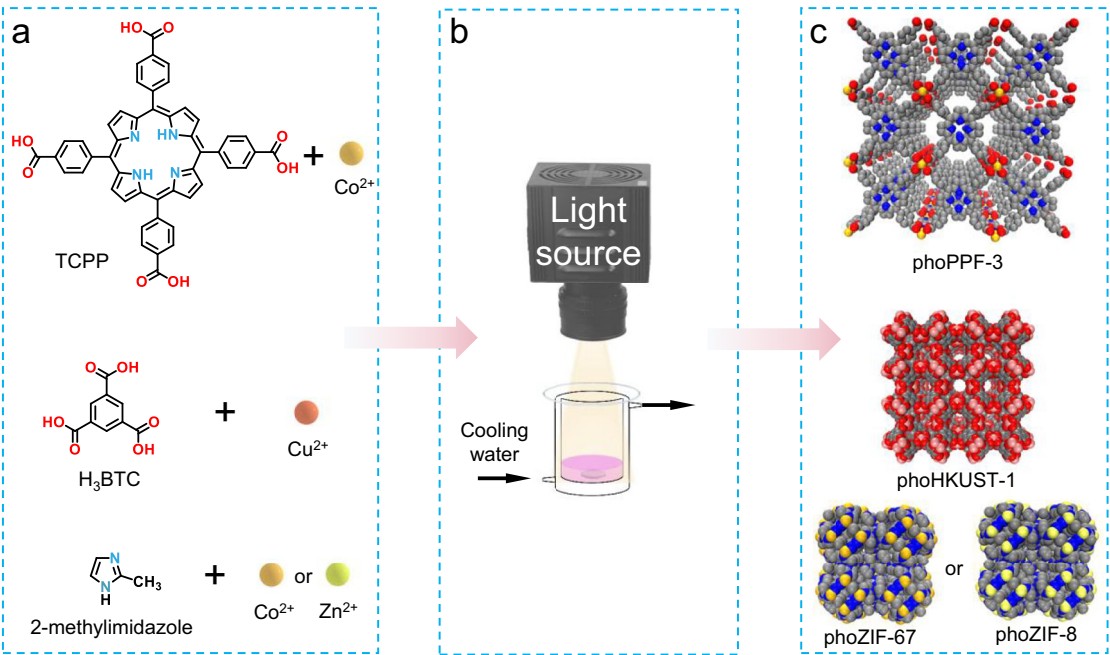

**Fig. 1 | Photosynthesis experimental setup, studied ligands and resulting MOF structures. a** Ligands of TCPP, $H_3$BTC, and 2-methylimidazole and metal ions of $Co^{2+}$, $Cu^{2+}$, and $Zn^{2+}$ used for MOF synthesis. **b** Schematic diagram of the photosynthesis setup. **c** Synthesized MOFs with their corresponding designations and structural illustrations (C, grey; N, blue; O, red).

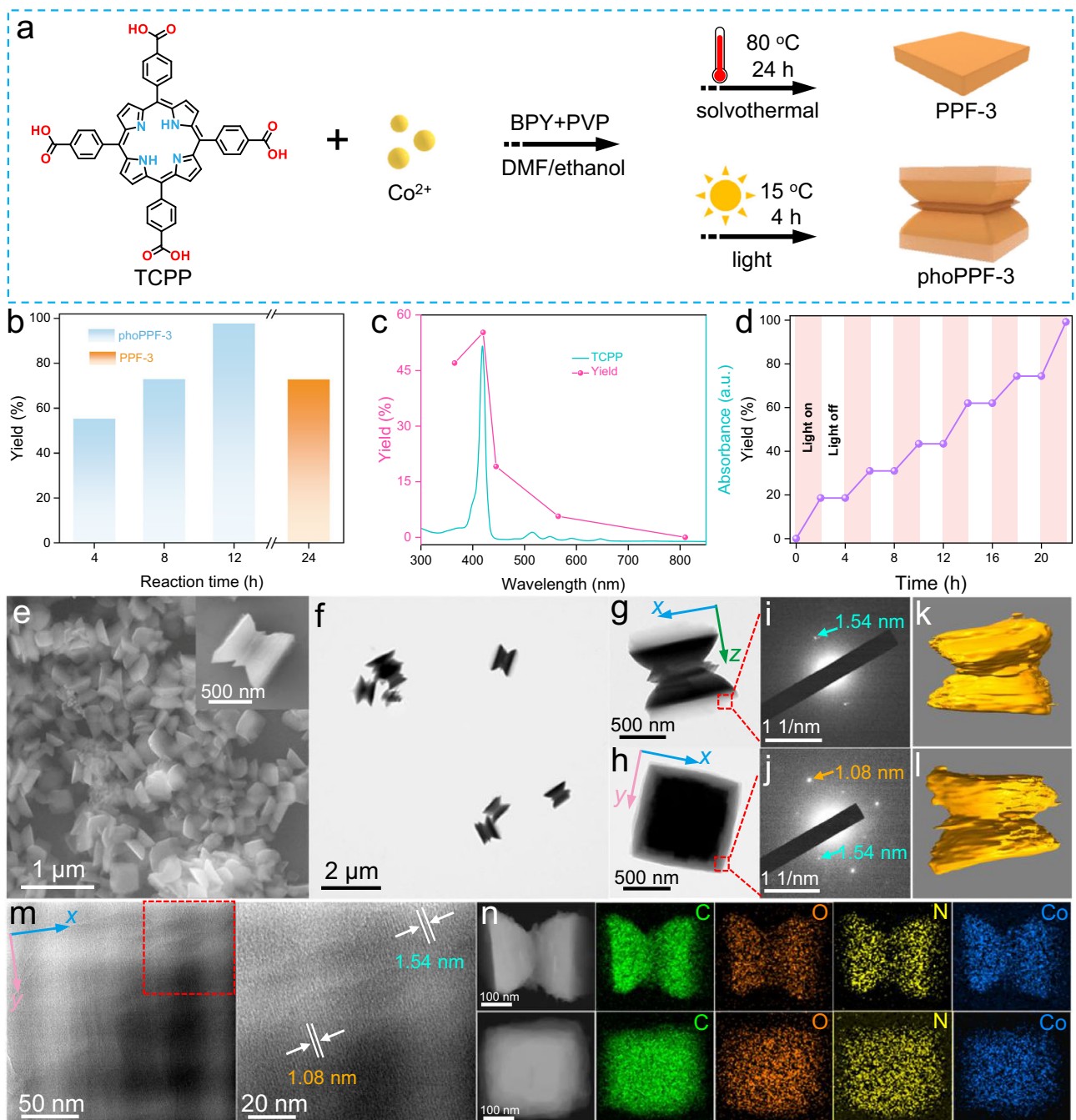

**Fig. 2 | Synthesis scheme and structural analysis of phoPPF-3. a** Scheme of the typical solvothermal synthesis of PPF-3 and the visible light-driven synthesis of phoPPF-3. **b** Time-dependent yield variation of phoPPF-3 under 420 nm LED light and the 24 h yield of PPF-3 using traditional solvothermal synthesis. **c** The 4 h yield of phoPPF-3 as a function of wavelength and UV-Vis spectrum of TCPP. **d** Time-dependent yield variation of phoPPF-3 under light on and light off conditions. Test conditions: the reaction substrate was three times the standard amount, and 2 mL of reaction solution was taken every 2 h for yield determination. Meanwhile, the light source was cycled between 2 h of light on and 2 h of light off. **e** SEM image and **f** TEM image of phoPPF-3. **g**, **h** TEM images and corresponding **i**, **j** SAED patterns (red squares shown in **g**, **h** of phoPPF-3, imaged along the side (x/z) and quasi-square end face (x/y) directions of the hourglass). The arrows marked in **i**, **j** have d-spacing of 1.54 (golden yellow) and 1.08 (cyan) nm. (**k**, **l**) 3D electron tomography reconstruction models of phoPPF-3 from different perspectives. **m** HRTEM image and corresponding zoomed-in region (red square) of phoPPF-3, imaged along the quasi-square end face edge (x/y direction) of the hourglass. The white arrows marked in **m** have d-spacing of 1.54 (golden yellow) and 1.08 (cyan) nm. **n** HAADF-STEM images and corresponding EDS mapping images of phoPPF-3, imaged along the side (top) and quasi-square end face (bottom) directions of the hourglass. Inset (red square) in **e** is SEM image of single phoPPF-3 particle. The blue, pink, and green arrows marked in **g**, **h**, and **m** are the x, y, and z directions, respectively.

shaped phoPPF-3 (Supplementary Figs. 1c and 2). As predicted, no product was obtained in the absence of light (Supplementary Fig. 1d). Under 420 nm LED light, phoPPF-3 achieved a high yield of 72.8% within 8 h, comparable to that obtained via the solvothermal method after 24 h (Fig. 2b). Figure 2c shows the 4 h-synthesis yields of phoPPF-

3 under LED lights at different wavelengths of 365, 420, 445, 565, and 810 nm, with the highest yield exceeding 55.0% at 420 nm. It is due to TCPP ligands having the strongest light absorption at 418 nm. Moreover, this light-driven synthesis exhibited excellent on/off control through light switching, enabling precise modulation of reaction

progression (Fig. 2d). Unlike conventional thermodynamically controlled processes, this system is kinetically controlled. Notably, the light-driven approach offers significant advantages, including lower temperature operation, shorter synthesis time, and potential compatibility with sunlight.

The morphology and composition of the as-prepared phoPPF-3 were characterized by scanning electron microscopy (SEM) and transmission electron microscopy (TEM). As shown in Fig. 2e, SEM imaging revealed an hourglass-shaped morphology characterized by quadrilateral faces at both ends and a layered structure in the central region. The side length (z-direction thickness) of the hourglass-like particles ranged from ~200 nm to ~600 nm, with the average particle thickness of ~352 nm, as determined by statistical analyses of the SEM image (Supplementary Fig. 3). TEM analysis confirmed this architecture while revealing delicate structural variations - the terminal ends exhibited less dense packing compared to the central region (Fig. 2f–h). We then focused on the light-contrast regions by using selected area electron diffraction (SAED), which suggests they are crystalline. The SAED pattern acquired perpendicular to the x/z plane showed a lattice spacing of 1.54 nm corresponding to the (100) crystal plane of the orthogonal ordered structure, while the pattern obtained perpendicular to the x/y plane exhibited two distinct spacings of 1.54 nm and 1.08 nm (attributed to the (001) plane) (Fig. 2i, j). In contrast, PPF-3 exhibited a nanosheet structure with a size of approximately 800 nm (Supplementary Fig. 4a–c). Notably, the SAED pattern reveals that PPF-3 shares the same lattice spacing as the end face (i.e., x/y plane) of phoPPF-3, indicating that it has a similar structural feature to the end face part of phoPPF-3 (Supplementary Fig. 4d). 3D topography reconstruction also validated the hourglass-shaped layered stacking structure of phoPPF-3 (Fig. 2k, l). Lattice spacing analysis of the end face edge (x/y-direction) further reveals two lattice spacings of 1.54 and 1.08 nm (Fig. 2m), which are consistent with SAED observations (Fig. 2h, j). High-angle annular dark-field scanning transmission electron microscopy (HAADF-STEM) imaging and accompanying energy-dispersive X-ray spectroscopy (EDS) elemental mapping, performed along the thickness direction and perpendicular to the end face of phoPPF-3, demonstrated an overall homogeneous distribution of C, O, N, and Co throughout the entire structure (Fig. 2n). A notable exception was observed at two terminal end regions, where all the elemental signals showed reduced intensities. This observation is in fully consistent with TEM results. Such difference in intensities at the ends and in the central region may cause variations in the band structure[12].

## Composition, optical property, and crystal structure of phoPPF-3

To elucidate the structural and compositional differences between phoPPF-3 and traditionally synthesized PPF-3, we performed X-ray diffraction (XRD) analysis, X-ray photoelectron spectroscopy (XPS), Fourier transform infrared spectroscopy (FTIR), and UV-Vis diffuse reflectance spectroscopy (UV-Vis DRS). The XRD patterns of phoPPF-3 and PPF-3 match closely, confirming their isostructural nature (Fig. 3a). However, phoPPF-3 exhibited a slightly higher degree of crystallinity, further demonstrating the superiority of this approach. Moreover, the high consistency between experimental and simulated XRD patterns of each sample confirms their crystal structure and further, their subtle crystal structure differences, between which phoPPF-3 only exhibited selective $Co^{2+}$-carboxylate coordination behavior, while normal PPF-3 showed the Co coordination with both carboxylate groups and nitrogen atoms. (Supplementary Figs. 5–7).

In the XPS analysis, the resolved C 1s peaks at 284.6, 285.8, and 287.7 eV are attributed to C=C/C−C/C−H, C−N/C−O, and C=O bonds, respectively (Supplementary Fig. 8). Compared with the C 1s spectra of PPF-3, the ratio of C=O to C 1s is higher in phoPPF-3, potentially due to the presence of partially uncoordinated carboxyl groups of TCPP

during visible light-driven synthesis (Supplementary Table 1). In the O 1s spectra of phoPPF-3, the peak at 532.6 eV, assigned to the −OH group, is considerably higher in that in PPF-3, indicating that some carboxyl groups in TCPP did not coordinate with Co ions (Fig. 3b). Interestingly, the retention of the unprotonated N (−C=N−) signal and the disappearance of the N−Co signal in the N 1s spectrum of phoPPF-3 as compared to PPF-3 suggest the lack of coordination between pyrrolic N and Co ions in phoPPF-3, implying that Co doesn't bind at the porphyrin center (Fig. 3c and Supplementary Fig. 9)[40–42]. The XPS Co 2p spectra of both phoPPF-3 and PPF-3 display characteristic peaks at ~781.4 (Co $2p_{3/2}$) and ~796.9 eV ($2p_{1/2}$), corresponding to $Co^{2+}$ (Fig. 3d)[43]. Additionally, two broad satellite peaks appear at ~784.8 and ~802.2 eV. These results support the presence of weak coordination in phoPPF-3. The FTIR results further corroborate the weak coordination of phoPPF-3, evidenced by the presence of a N−H in-plane bending mode at 969 cm$^{-1}$ and a N−H stretching vibration at 3315 cm$^{-1}$, both of which are located within the porphyrin center, and a high content of the C=O stretching vibration at 1653 cm$^{-1}$, which suggests the formation of $Co_2(COO)_4$ paddlewheel unit cells (Fig. 3e and Supplementary Fig. 10)[40]. Furthermore, the absence of an absorption peak at 1005 cm$^{-1}$ suggests that the hydrogen proton was not substituted by the cobalt ion (N−Co), further indicating that the Co ions are not coordinated at the porphyrin center[44]. In addition, the Raman spectrum of phoPPF-3 showed a high degree of similarity to that of TCPP (Supplementary Fig. 11).

UV-Vis DRS was further performed to probe the central coordination of TCPP. The UV-Vis DRS spectrum of phoPPF-3 showed four absorption bands at 517, 553, 592, and 649 nm, corresponding to the Q bands of the porphyrin unit (Fig. 3f and Supplementary Fig. 12). This further indicates that Co ions are not coordinated at the porphyrin center within phoPPF-3[40]. Conversely, a single Q band at 540 nm in PPF-3 corroborates Co ions' coordination at the porphyrin center. This phenomenon was also confirmed by the UV-Vis spectrum of PPF-3 dispersed in solution (Supplementary Fig. 12). In addition, the Soret band of phoPPF-3 in solution was split and red-shifted, appearing at 421 and 431 nm compared to the Soret band of TCPP at 415 nm, which further confirming the MOF formation (Supplementary Fig. 12)[40]. Moreover, PPF-3 exhibits strong J-aggregation at 433 nm, along with its relatively thinner, sheet-like morphology. In contrast, phoPPF-3 displays weak J-aggregation at 430 nm and a thicker morphology. Therefore, the phoPPF-3 with Co-free coordination in porphyrin center processed a weak charge repulsion between the MOF layers, which promotes H-stacking of phoPPF-3 units[40].

Based on all the above characterization results, the crystal structural models of phoPPF-3 and PPF-3 are proposed (Fig. 3g). In PPF-3, the Co atoms coordinate not only at the porphyrin centers but also within the paddlewheel units. In contrast, phoPPF-3 primarily features Co atoms coordinated in the paddlewheels. Therefore, TCPP ligands coordinate with Co metal nodes under varying conditions, forming a slightly varied 2D chessboard-like sheets (Fig. 3g). These two different sheet structures are further pillared by BPY molecules in AB and AA stacking modes, respectively, ultimately yielding the final 3D network structures of PPF-3 and phoPPF-3. The Co-free coordination in porphyrin center, coupled with BPY acting as a pillar that coordinates to the $Co^{2+}$ ions of the paddle-wheel nodes, promotes an AA stacking pattern of phoPPH-3 due to strong coordination force between functional groups and metal nodes. Additionally, the Co content in phoPPF-3 (9.97 wt%) is slightly lower than that in PPF-3 (11.54 wt%). These results closely match the theoretical values calculated for the proposed unit structures (12.36 wt% for PPF-3 unit (Co:TCPP:BPY = 3:1:3) and 9.68 wt% for phoPPF-3 unit (Co:TCPP:BPY = 2:1:2)). Moreover, the TCPP to BPY molar ratios, quantified via $^1H$ NMR as 0.41 for PPF-3 and 0.58 for phoPPF-3, provide direct stoichiometric validation for our proposed connectivity scheme (Supplementary Fig. 13). The metal-specific coordination behavior of phoPPF-3 underscores the distinct

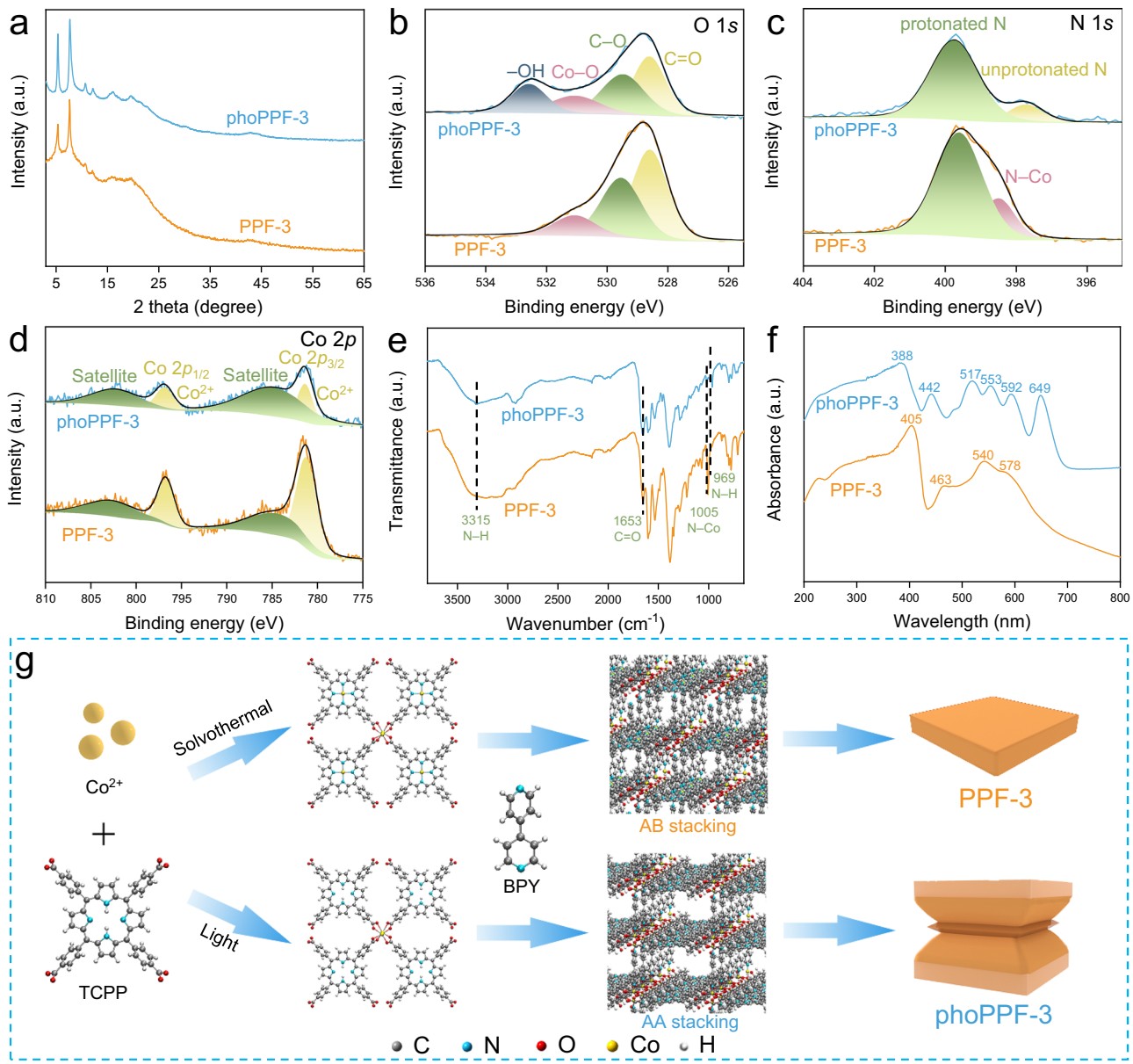

**Fig. 3 | Systematic structural and compositional characterization of phoPPF-3.** **a** XRD patterns and XPS spectra of **b** O 1s, **c** N 1s, and **d** Co 2p of phoPPF-3 and PPF-3. **e** FTIR spectra and **f** UV-Vis DRS spectra of phoPPF-3 and PPF-3. **g** Schematic of PPF- 3 and phoPPF-3 molecular assembly, depicting the formation of layered unit structures and 3D architectures with AA and AB stacking modes.

advantages of the light-driven synthesis approach, offering an opportunity for MOF optimization and utilization.

### Thermal stability of phoPPF-3

The stability of the as-synthesized phoPPF-3 was systematically investigated and compared with conventional alternatives. The phoPPF-3 can be well-dispersed in solvent after shaking and does not show any change up to 20 h, which is the longest observation time under the current investigation (Supplementary Fig. 14). Moreover, the thermal stability of as-prepared phoPPF-3 was investigated by thermogravimetric analysis (TGA) and TG/DTA-MS (Fig. 4a–c). A 2.0% of weight loss was observed for both phoPPF-3 and PPF-3 at 200 °C under argon, which can be attributed to the loss of absorbed water and DMF (Fig. 4b, c). When the temperature increases to 300 °C, there is about 7.0% and 8.5% of weight loss for phoPPF-3 and PPF-3, respectively. This loss was mainly caused by the thermal-induced amorphization, which may lead

to a partial collapse of the paddlewheel structure (Fig. 4b, c). In the temperature range of 210–415 °C, phoPPF-3 demonstrated marginally enhanced thermal stability compared to PPF-3. This stability trend reversed above 415 °C, which we attribute to the decomposition of paddlewheel structures at a higher content in phoPPF-3. Notably, phoPPF-3 exhibited a 34 °C higher decomposition temperature (418.5 °C) than PPF-3 (384.5 °C), estimated from the DTA curves (Supplementary Fig. 15). These results demonstrate the higher thermal stability of phoPPF-3 compared to PPF-3 below 415 °C, highlighting its promise for long-term applications in a reasonably high temperature operation window. To further probe their thermal stability, we conducted in situ TEM studies using laser-induced heating (Fig. 4d). phoPPF-3 maintained its structural integrity even at elevated laser fluences, demonstrating its high thermal stability (Fig. 4e), while PPF-3 underwent significant morphological changes, forming small nanoparticles on the MOF surface (Fig. 4f). These findings unequivocally

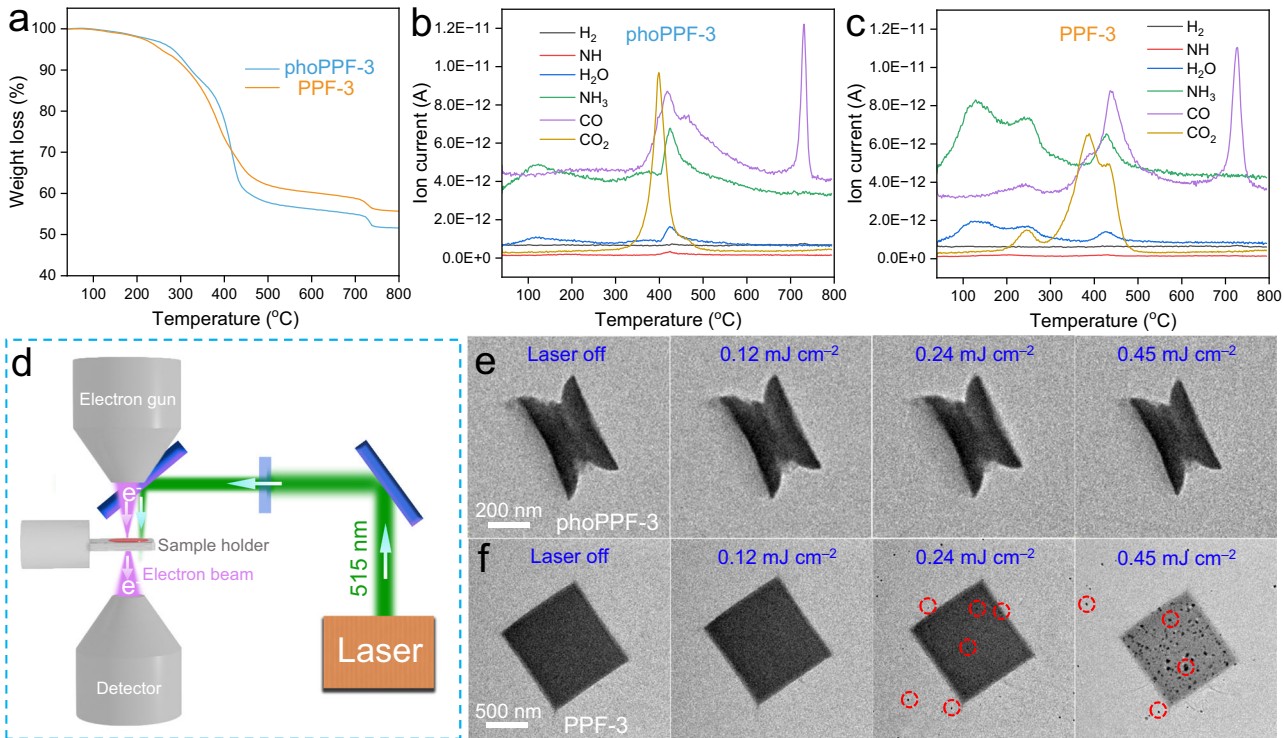

**Fig. 4 | Thermal stability analysis of phoPPF-3. a** TGA curves of phoPPF-3 and PPF-3 obtained under an argon atmosphere. TG/DTA-MS curves of **b** phoPPF-3 and **c** PPF-3 obtained under an argon atmosphere. **d** Schematic diagram of the in situ laser heating setup within a laser-integrated TEM. The arrows marked in **d** are the directions of laser and electron propagation. TEM images of **e** phoPPF-3 and **f** PPF-3 irradiated with varying laser fluences. The red dashed circles in **f** represent the aggregated nanoparticles.

demonstrate the enhanced thermal stability of phoPPF-3 relative to that of PPF-3.

## Formation mechanism of phoPPF-3

To uncover the growth kinetics, we employed TEM analysis to monitor time-dependent morphological evolution of phoPPF-3 across the critical reaction time window (Fig. 5a–d). Within 10 min, layered stacked square columnar structures emerged, driven by strong π - π stacking interaction between Co-TCPP molecules, facilitated by their conjugated π - electron systems. By 30 min, the initial sandwich-like architecture formed, where light exposure preferentially activated the terminal regions, accelerating molecular stacking. Further irradiation (90 min) produced a well-defined hourglass-like phoPPF-3, with the final hourglass-like MOF crystallizing after 4 h. Notably, the edge size at both ends expanded significantly due to rapid growth kinetics, while the central region developed more slowly. Size evolution analysis (Fig. 5e) confirmed anisotropic growth, with the thickness-direction growth rate exceeding both the lateral growth at the end face and the central region, highlighting differential reaction kinetics across the structure. Moreover, the increasing contrast of the top view of phoPPF-3 over time further indicates the gradual increase in the thickness.

The underlying microscopic mechanisms and solvation structures of Co-MOF synthesized under these two different conditions were further investigated through molecular dynamics (MD) simulations, following established methodologies (Supplementary Data 1–4)[45,46]. Supplementary Fig. 16a–c illustrates the representative snapshots of the binding process of phoPPF-3 during the simulation. Initially ($t = 0$ ns), all components, including solvent molecules, BPY, PVP, and phoPPF-3 units, were uniformly dispersed in the simulated box. By $t = 10$ ns, extensive self-assembly of phoPPF-3 units was observed, with solvent molecules, BPY, and PVP accumulating at the surface of the forming phoPPF-3 self-assembly. At equilibrium, phoPPF-3 units completed their self-assembly,

resulting in a loosely packed structure. In contrast, when $t = 10$ ns, PPF-3 units quickly aggregated into smaller, discrete assemblies, which later coalesced into a single, densely packed structure (Supplementary Fig. 16d–f). Figure 5f–i highlight that BPY molecules are mainly localized near the $Co^{2+}$-carboxylate coordination sites in phoPPF-3, corroborating the proposed 3D radial growth mechanism.

To investigate self-assemblies of Co-MOF, we analyzed the radial distribution functions (RDF) and coordination numbers (CN) derived from MD simulations. The RDF represents the probability of locating a particle at a specific distance ($r$) from a reference particle, revealing local structural order and intermolecular interactions. As shown in Fig. 5j, k, and n, the RDF peaks of PPF-3 are at 0.153, 0.183, 0.197, 0.259, 0.383, and 0.427 nm, which are shorter compared to those of phoPPF-3 (0.159, 0.196, 0.261, 0.264, 0.386, and 0.438 nm), suggesting a more compact structure in PPF-3. In addition, the CN, derived by integrating the first RDF peak, reflects the number of neighboring atoms or molecules surrounding a central atom within a defined cutoff distance, offering insights into local atomic packing, bonding environments, and solvation structures. The CN of PPF-3 (1.63, 2.19, 2.35, 5.63, 14.17, and 17.76) at the corresponding RDF peaks is slightly higher than that of phoPPF-3 (1.61, 2.07, 2.31, 5.62, 13.66, and 16.75), providing additional evidence for the thermally induced densification of the MOF framework (Fig. 5l, m and Supplementary Table 2). Therefore, the looser structure and more BPY molecular support lead to the growth of MOFs towards 3D structures, while solvothermal condition makes it easier for MOFs to form 2D layered structures.

The coordination structural differences between phoPPF-3 and PPF-3 may arise from the distinct electronic configurations of TCPP in its ground and excited states. To elucidate the photoinduced effects on TCPP's electronic structure, we performed density functional theory (DFT) calculations using Gaussian (Supplementary Data 5 and 6). The results revealed significant changes upon photoexcitation: the

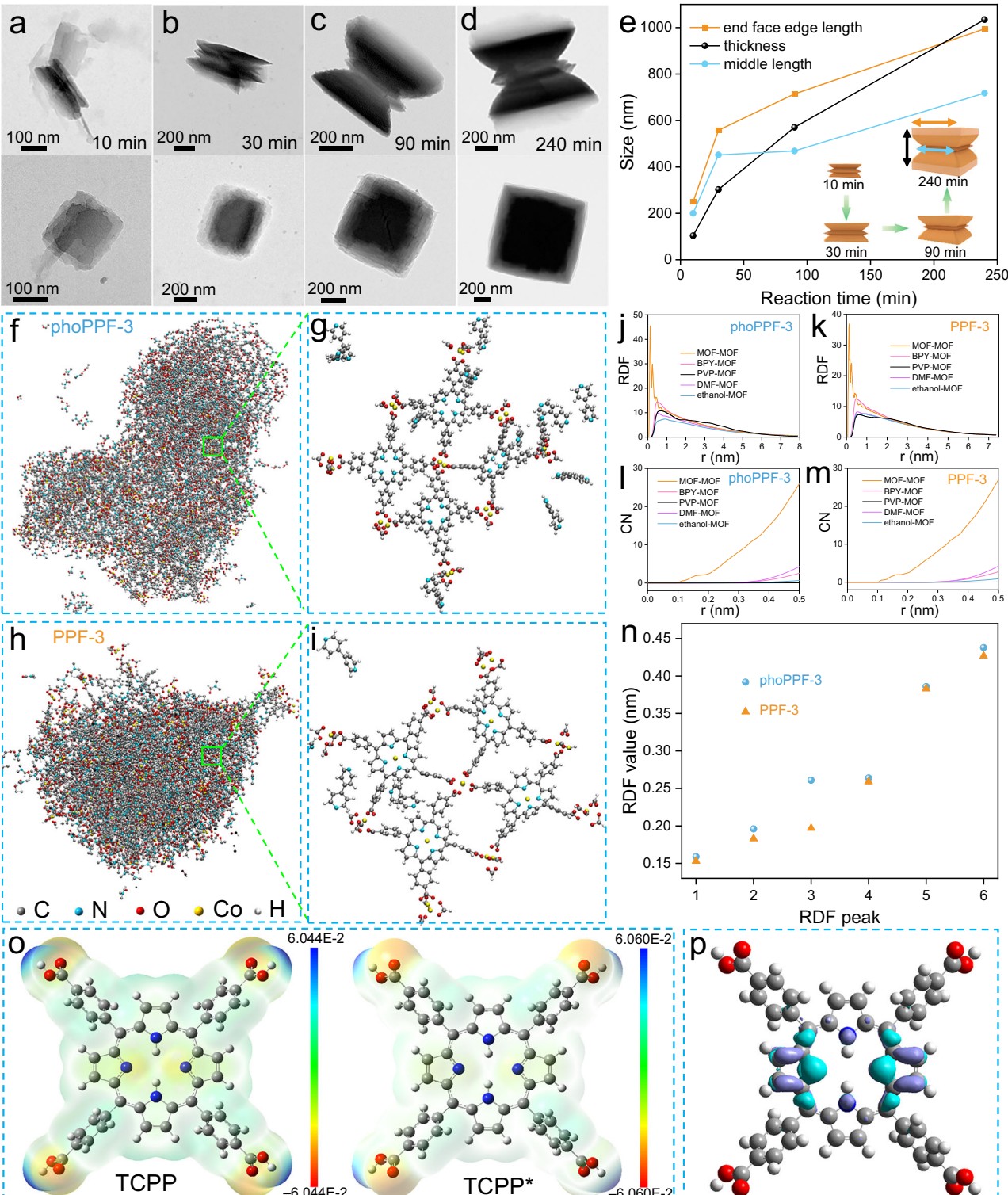

**Fig. 5 | Time dependent growth and MD simulation of the hourglass-shaped phoPPF-3.** TEM images of the side and end face of phoPPF-3 obtained at different reaction times of **a** 10 min, **b** 30 min, **c** 90 min, and **d** 240 min. **e** Time dependent size change of phoPPF-3. The inset in **e** shows the schematic of phoPPF-3 growth over time (green arrows). The black, light blue, and brown arrows are the thickness, middle length, and end face edge length, respectively. Snapshots of the MD simulated box of phoPPF-3 **f** and PPF-3 **h** in ball-stick representation, and

corresponding solvation structures **g**, **i** (the enlargement of green squares in **f**, **h**). Radial distribution function and coordination number of different components in phoPPF-3 (**j**, **l**) and PPF-3 (**k**, **m**). **n** The comparison of different RDF peaks in phoPPF-3 and PPF-3. **o** Electron density profiles of TCPP and TCPP* (C, grey; N, blue; O, red; H, white). **p** Difference in electron density between the excited and ground states of TCPP.

dipole moment decreased from 2.279 D (ground state) to 2.039 D (excited state), while the polarizability increased from 766.78 a.u. to 848.02 a.u. (Supplementary Table 3). This electron density redistribution within the porphyrin ring is visualized in Fig. 5o, p, particularly around the C and N atoms, as well as a pronounced charge transfer evident in the electron density difference map (Fig. 5p). These electronic structure variations between TCPP and TCPP* likely modulate their coordination behavior with $Co^{2+}$ ions, thereby contributing to the observed coordination structural divergence between phoPPF-3 and PPF-3. Specifically, this photoinduced electronic modification in TCPP* reduces the electron density or alters the accessibility of the porphyrin nitrogen atoms, making them less favorable for $Co^{2+}$ coordination during the MOF formation. In contrast, in PPF-3, $Co^{2+}$ preferentially coordinates to the porphyrin nitrogens of TCPP, leading to the formation of metalloporphyrin units. Furthermore, orbital coupling of layered structures in their electron-excited states can influence anisotropic growth rates, yielding layered morphologies[33]. This difference in preferred coordination geometry and binding sites directly determined the resulting MOF growth pattern. However, it is important to note that these computational approaches are subject to limitations, as accurately simulating the exact state of the materials under dynamic operating conditions remains a significant challenge.

The influence of PVP and BPY on the morphology of light-induced phoPPF-3 was systematically investigated. In the absence of PVP, the resulting phoPPF-3$_{(no\ PVP)}$ exhibited a long nanoribbon morphology (Supplementary Fig. 17a). While the sample retained a multilayered structure, J-aggregation was enhanced in the PVP-free system, driving 1D assembly (Supplementary Fig. 17b, c). Moreover, the fast Fourier-transform (FFT) analysis revealed larger lattice spacing in phoPPF-3$_{(no\ PVP)}$ compared to phoPPF-3 or PPF-3 (Supplementary Fig. 17d), suggesting weaker interlayer interactions. In addition, without BPY, phoPPF-3$_{(no\ BPY)}$ formed ultrathin, irregular 2D nanosheets, confirming BPY's role as a face-to-face molecule linker for Co-TCPP coordination (Supplementary Fig. 18a–c). Moreover, the polycrystalline nature of these nanosheets was evidenced by FFT patterns, with lattice spacings comparable to PhoPPF-3$_{(no\ PVP)}$ (Supplementary Fig. 18d). Furthermore, in the absence of both BPY and PVP, the product formed irregular stacked bilayers with nonuniform sizes (Supplementary Fig. 19). These results fully demonstrate the important influence of PVP and BPY on morphology control and Co-TCPP assembly.

The effect of reaction temperature on the morphology of phoPPF-3 was further investigated by synthesizing MOFs at different temperatures (0, 5, 10, 15, 20, and 25 °C) under 420 nm LED illumination and analyzed the products using SEM (Supplementary Fig. 20). At lower temperatures (0–5 °C), phoPPF-3 formed interconnected fibrous filaments. As the temperature increased to 10 °C, 3D MOF structures emerged alongside fiber-like assemblies. At 15 °C, the product exhibited a regular hourglass morphology, while temperature above 20 °C yielded cubic structures. These findings highlight the pronounced influence of reaction temperature on the morphology of phoPPF-3. The morphology variations can be attributed to temperature-dependent reactant diffusion rates, which modulate anisotropic growth kinetics and ultimately dictate the final morphology.

To evaluate the contribution of light in this visible light-driven synthesis strategy, we analyzed phoPPF-3 synthesized under lights of different wavelengths (365, 495, and 595 nm) and also different light sources (LED and Xenon lights). Under the irradiation of 365, 495, and 595 nm LED lights, the obtained phoPPF-3 consistently showed the hourglass-like morphology, demonstrating the broad spectral responsiveness of the light-driven synthesis strategy (Supplementary Fig. 21–23). Furthermore, when a 300 W Xenon lamp spanning 200 to 2500 nm was employed as an alternative light source, the same morphology was successful achieved (Supplementary Fig. 24). The product yield exhibited a strong dependence on light power (Supplementary Fig. 25). Increasing the light power density from 780.3 to 2191.1 W m$^{-2}$

significantly enhanced the phoPPF-3 yield from 22.2% to 42.9%. This linear response suggests that the photochemical synthesis is primarily driven by a photon-governed conversion process. To further validate the practicality of this strategy, we replaced the LED with a solar simulator (standard sunlight: 1000 W m$^{-2}$), which similarly yield hourglass-shaped phoPPF-3 (Supplementary Fig. 26). Notably, we successfully achieved direct outdoor synthesis of phoPPF-3 with a yield of 20.2% using natural sunlight, a readily accessible and sustainable light source (Supplementary Fig. 27). The successful synthesis of phoPPF-3 under sunlight was further confirmed by UV-Vis absorption spectroscopy (Supplementary Fig. 28). These results confirm that the light-driven synthesis strategy is not only versatile across different wavelengths but also compatible with diverse light sources, highlighting its potential for scalable and sustainable MOF production.

## Universality of the light-driven MOF synthesis

To further demonstrate the broad applicability of the light-driven synthesis approach, we systematically evaluated three representative MOF systems incorporating distinct metal nodes and organic likers: (i) HKUST-1 ($Cu^{2+}$), (ii) ZIF-67 ($Co^{2+}$), and (iii) ZIF-8 ($Zn^{2+}$). The morphology and composition of MOFs prepared by light-driven synthesis (420 nm LED light, 15 °C, 4 h) were characterized by TEM and XRD. As shown in Fig. 6a–f, phoHKUST-1, phoZIF-67, and phoZIF-8 showed the similar morphologies compared to HKUST-1 (110 °C, 24 h), ZIF-67 (25 °C, 24 h), and ZIF-8 (25 °C, 24 h). Furthermore, the XRD results confirm the formation of all target MOFs with well-defined crystalline structure (Fig. 6g–i). The simulation results aligned well with the experimental patterns, further confirming the successful light-driven synthesis of HKUST-1, ZIF-67, and ZIF-8 (Supplementary Figs. 29–31). The synthesis of these MOFs involves different precursors, ligands, and reaction mechanisms that do not rely on photoexcitation of a specific ligand to control coordination behavior or morphology. Light may be absorbed or scattered by MOF seed crystals, thereby affecting their further growth (Supplementary Fig. 32). Furthermore, we also synthesized phoMn-TCPP by substituting $Mn^{2+}$ for $Co^{2+}$ in the photochemical synthesis of phoPPF-3. For phoMn-TCPP, the UV-Vis spectral changes and the time-dependent yield variation under light on and light off conditions provided direct evidence for a photosynthesis process driven by TCPP photoexcitation (Supplementary Fig. 33). Therefore, this visible light-driven protocol offers significant advantages over conventional solvothermal methods, such as time saving, mild conditions, and energy efficiency. The successful preparation of these structurally diverse MOFs (especially composed of ligands with photoexcitation effects) robustly validates the generality of the photochemical strategy for coordination polymer synthesis.

## Photocatalytic performance of phoPPF-3

The photocatalytic performance of phoPPF-3 was evaluated for the oxidation of benzyl alcohol in acetonitrile solution using carbon dioxide as the oxidant under light irradiation (300 W Xenon Lamp). As shown in Fig. 7a, the photocatalytic performance of phoPPF-3 and PPF-3 was evaluated by quantifying the benzaldehyde yield. After 12 h of reaction, the benzaldehyde yields of 10.15% and 13.49% were obtained with PPF-3 and phoPPF-3, respectively (Supplementary Table 4). Notably, phoPPF-3 showed a significant ~32.9% increase in catalytic activity over PPF-3. To further validate the enhanced photocatalytic performance of phoPPF-3, we investigated photocatalytic $H_2$ evolution of PPF-3 and phoPPF-3 under Xenon lamp irradiation. Following 12 h of irradiation, phoPPF-3 achieved a $H_2$ rate of 557.6 μmol g$^{-1}$ h$^{-1}$, while PPF-3 showed no detectable $H_2$ production under identical conditions (Supplementary Figs. 34–37). Photocurrent measurements revealed slightly higher responses for phoPPF-3 (-0.36 μA cm$^{-2}$) compared to PPF-3 (-0.33 μA cm$^{-2}$), suggesting enhanced light-harvesting capability and/or photogenerated carrier generation efficiency in phoPPF-3 (Fig. 7b). However, electrochemical impedance spectroscopy (EIS)

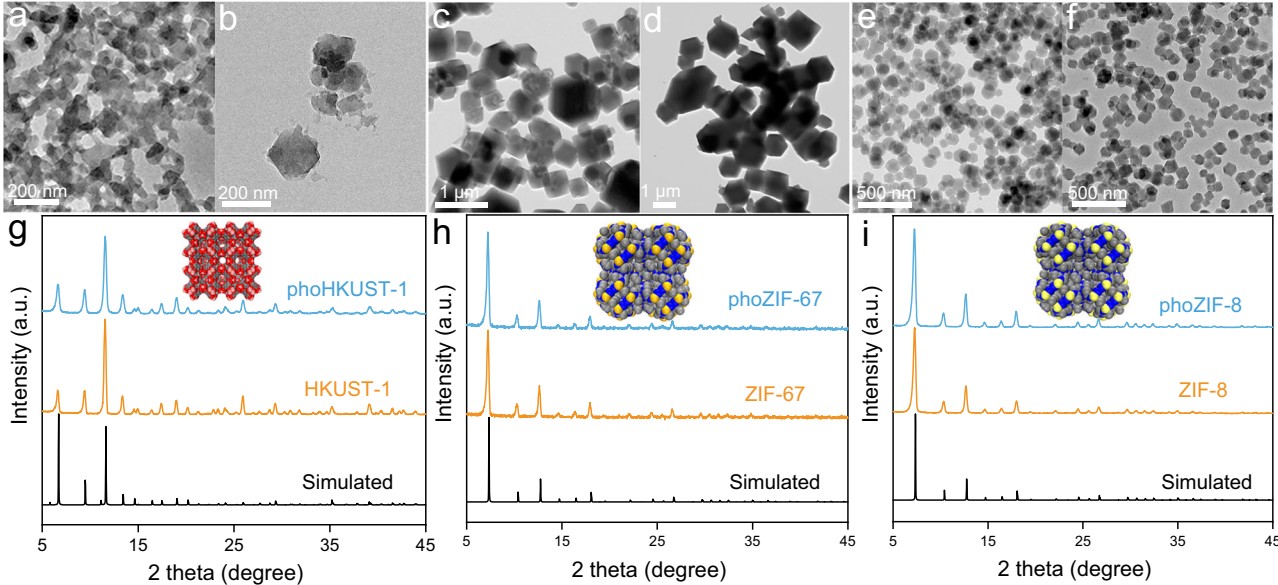

**Fig. 6 | Universality of light-driven MOF synthesis.** TEM images of **a** phoHKUST-1 and **b** HKUST-1, **c** phoZIF-67 and **d** ZIF-67, and **e** phoZIF-8 and **f** ZIF-8. Experimental and simulated XRD patterns of **g** phoHKUST-1 and HKUST-1, **h** phoZIF-67 and ZIF-67, and **i** phoZIF-8 and ZIF-8. Insets in **g**–**i** are the corresponding modelled crystal structures (C, grey; N, blue; O, red; Cu, brownish red; Co, golden yellow; Zn, pale yellow).

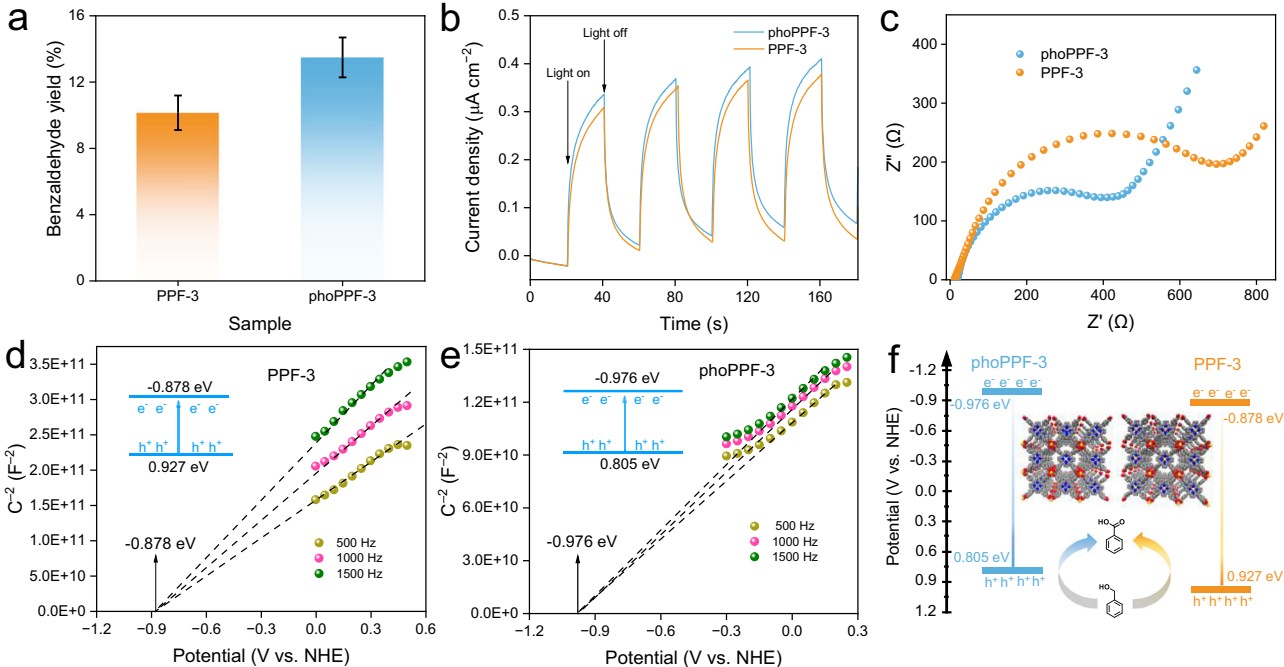

**Fig. 7 | Photocatalytic performance of phoPPF-3. a** Benzaldehyde yield of PPF-3 and phoPPF-3 under the following photocatalytic conditions: 25 °C, 300 W Xenon lamp, 10 mg of photocatalyst, the benzyl alcohol concentration of 4.82 mM, and the reaction time of 12 h. The error bars indicate one standard deviation based on three measurements. **b** Transient photocurrent responses of phoPPF-3 under intermittent illumination. **c** EIS Nyquist plots of PPF-3 and phoPPF-3. Mott-Schottky plots of **d** PPF-3 and **e** phoPPF-3. The insets in **d** and **e** show the conduction band and valence band positions of PPF-3 and phoPPF-3, respectively. **f** Schematic illustration of the photocatalytic mechanism of PPF-3 and phoPPF-3. Insets in **f** are the corresponding modelled crystal structures (C, grey; N, blue; O, red; Co, golden yellow).

demonstrated a smaller semicircle for phoPPF-3 than for PPF-3, indicating a lower interfacial charge transfer resistance of phoPPF-3 (Fig. 7c). Therefore, the enhanced photocatalytic efficiency of phoPPF-3 is likely attributed to Co metal vacancies, which facilitate PCET process via hydrogen atoms in the porphyrin framework[47,48]. This PCET process directly provides both electrons and protons to the reactants, thereby accelerating the reaction rate. In addition, compared to PPF-3, the increased surface area (1.67-fold), pore size (1.53-fold), and pore volume (2.54-fold) of phoPPF-3 are likely to act synergistically with the intrinsically active sites, facilitating their more effective utilization and consequently enhancing photocatalytic performance (Supplementary Fig. 38 and Supplementary Table 5).

To investigate the electronic structural properties of phoPPF-3 and PPF-3, Mott-Schottky measurements were conducted at frequencies of 500, 1000, and 1500 Hz. Both of PPF-3 and phoPPF-3 exhibited positive slopes, characteristic of n-type semiconductors, with flat-band potentials of −0.878 and −0.976 eV, respectively (Fig. 7d, e). The conduction and valence band positions of PPF-3 and phoPPF-3, calculated based on Tauc plots (Supplementary Fig. 39) and Mott-Schottky analysis (Fig. 7f), confirmed that both photocatalysts meet the thermodynamic requirements for photocatalytic oxidation of benzyl alcohol to benzaldehyde and $H_2$ evolution. Therefore, the light-driven synthetic approach yields phoPPF-3 with electronic structural property comparable to that of conventional, solvothermally-synthesized PPF-3, while offering advantages in energy efficiency, reaction time, and photocatalytic activity.

## Techno-economic analysis of photochemical MOF synthesis

To evaluate industrial potential, we conducted a techno-economic analysis (TEA) comparing photochemical synthesis with conventional solvothermal route, focusing on energy, environmental metrics (E-factor and process mass intensity (PMI)), and scalability. The photochemical route offers an estimated energy saving over the solvothermal method, a benefit amplified if sunlight can be utilized. Conversely, laboratory-scale analysis on the E-factor and PMI indicates lower apparent economic efficiency due to high solvent usage inherent to milligram-scale batch synthesis. Such challenges are typical for early-stage synthetic development.

Scalability testing on a milligram-scale (~2 mg per batch) revealed that doubling the precursor concentration increased yield, but a threefold increase did not. This suggests that the light penetration became a limiting factor, and higher concentration may have hindered conversion due to increased light absorption by the reaction mixture, causing considerable light attenuation. But this can be solved by using a larger light source to increase the illumination spot falling on the reaction solution. Further, considering the photochemical method operates under visible light and ambient conditions, which can be readily integrated into the continuous-flow photoreactor design, this method offers significant advantages for industrial scale-up. Thus, this TEA highlights the energy-saving benefits and practical potential of photochemical MOF synthesis (Supplementary Table 6), while also identifying key areas for optimization in waste reduction and reactor design to improve overall process sustainability and economy.

In summary, we have developed a visible light-driven strategy for the effective synthesis of MOFs that offers significant advantages over conventional solvothermal methods. This green, energy-efficient approach enables rapid synthesis while producing structurally materials, as exemplified by phoPPF-3 with its distinctive 3D hourglass-shaped morphology and metal-free porphyrin coordination center. The photochemically synthesized phoPPF-3 demonstrated enhanced thermal stability and photocatalytic performance (1.5 times increase in benzyl alcohol oxidation activity compared to solvothermally prepared 2D PPF-3 nanosheets). The practicality of methodology is evidenced by successful solar-driven synthesis and broad substrate applicability (including HKUST-1, ZIF-67, and ZIF-8). This work establishes a paradigm for energy-efficient materials design, combining reduced environmental impact with precise control over structural and functional properties. The demonstrated ability to generate materials with tailored architectures and enhanced functionalities through photochemical pathways opens an avenue for sustainable materials synthesis.

## Methods

### Synthesis of phoPPF-3

In separate vials, 78.0 mg of BPY, 4.4 mg of $Co(NO_3)_2 \cdot 6H_2O$, and 4.0 mg of TCPP were dissolved in 5 mL, 6 mL, and 2 mL of a 3:1 (v/v) DMF/ethanol mixture, respectively. Then, 100 μL of the BPY solution was added to the $Co(NO_3)_2$ solution, followed by sonication. Subsequently, 10.0 mg of PVP and the above-prepared

TCPP solution were added to the mixture of $Co(NO_3)_2$ and BPY. The resultant solution of $Co(NO_3)_2$/BPY/PVP/TCPP was transferred to a photocatalytic reactor, maintained at 15 °C, and stirred while being irradiated with a 100 W LED (420 nm) for 8 h. The power density of the light source irradiated on the sample was ~8 mW cm$^{-2}$. Finally, the product was collected by centrifugation, washed with ethanol, and re-dispersed in ethanol for further use. The synthesis was also done under different light sources with varying wavelengths by using the same procedure.

### Synthesis of PPF-3 nanosheets

The precursor mixture solution of $Co(NO_3)_2$/BPY/PVP/TCPP, prepared as described for phoPPF-3, was transferred to a sealed vial and subjected to stirring at 80 °C for 24 h. After the reaction, the product was collected by centrifugation, washed, and then dispersed in ethanol.

### Synthesis of HKUST-1

HKUST-1-H was synthesized using a modified literature procedure[49]. Briefly, 0.716 g of $Cu(NO_3)_2 \cdot 3H_2O$ and 0.421 g of trimesic acid were mixed in 12 mL of an ethanol/water ($v/v = 1$:1) solution in a beaker. The mixture was stirred for 30 min, transferred into a Teflon-lined autoclave[50] and heated at 110 °C for 24 h. After the reaction, the resulting product was collected by centrifugation. It was then redispersed in ethanol, centrifuged, and collected. This cycle was repeated three times. Finally, the purified product was dispersed in ethanol for further investigations.

### Synthesis of phoHKUST-1

In a typical synthesis, 0.716 g of $Cu(NO_3)_2 \cdot 3H_2O$ and 0.421 g of trimesic acid were mixed in 12 mL of an ethanol/water ($v/v = 1$:1) solution in a beaker. The mixture was then stirred for 30 min, transferred to a photocatalytic reactor maintained at 15 °C, and stirred while being irradiated with a 100 W LED (420 nm) for 4 h. Finally, the product was collected by centrifugation, washed with ethanol, and re-dispersed in ethanol for further use.

### Synthesis of ZIF-67

ZIF-67 was synthesized via a modification of a previously reported procedure[51,52]. Briefly, 0.388 g of $Co(NO_3)_2 \cdot 6H_2O$ and 0.411 g of 2-methylimidazole were introduced separately into 10 mL of methanol. After the complete dissolution, the two solutions were mixed and continuously stirred at ambient temperature for 24 h. The resultant precipitate was collected by centrifugation, washed with methanol, and dried at 60 °C.

### Synthesis of phoZIF-67

In a typical synthesis, 0.388 g of $Co(NO_3)_2 \cdot 6H_2O$ and 0.411 g of 2-methylimidazole were dissolved in 10 mL of methanol, respectively. The two solutions were then mixed, transferred to a photocatalytic reactor maintained at 15 °C, and stirred under irradiation with a 100 W LED (420 nm) for 4 h. The resulting product was collected by centrifugation, washed with methanol, and dried at 60 °C.

### Synthesis of ZIF-8

ZIF-8 was synthesized via a modification of a previously reported procedure[53]. Briefly, 0.397 g of $Zn(NO_3)_2 \cdot 6H_2O$ and 0.411 g of 2-methylimidazole were introduced separately into 10 mL methanol. After the complete dissolution, the two solutions were mixed and continuously stirred at ambient temperature for 24 h. The resultant precipitate was collected by centrifugation, washed with methanol, and dried at 60 °C.

### Synthesis of phoZIF-8

In a typical synthesis, 0.397 g of $Zn(NO_3)_2 \cdot 6H_2O$ and 0.411 g of 2-methylimidazole were dissolved in 10 mL of methanol, respectively. The two solutions were then mixed, transferred to a photocatalytic reactor maintained at 15 °C, and stirred under irradiation with a 100 W

LED (420 nm) for 4 h. The resulting product was collected by centrifugation, washed with methanol, and dried at 60 °C.

## Synthesis of phoMn-TCPP

In separate vials, 78.0 mg of BPY, 2.7 mg of $MnCl_2 \cdot H_2O$, and 4.0 mg of TCPP were dissolved in 5 mL, 6 mL, and 2 mL of a 3:1 ($v/v$) DMF/ethanol mixture, respectively. Then, 100 μL of the BPY solution was added to the $MnCl_2$ solution, followed by sonication. Subsequently, 10.0 mg of PVP and the above-prepared TCPP solution were added to the mixture of $MnCl_2$ and BPY. The resultant solution of $MnCl_2$/BPY/PVP/TCPP was transferred to a photocatalytic reactor, maintained at 15 °C, and stirred while being irradiated with a 100 W LED (420 nm) for 8 h. Finally, the product was collected by centrifugation, washed with ethanol, and redispersed in ethanol for further use.

## Synthesis of Mn-TCPP

The precursor mixture solution of $MnCl_2$/BPY/PVP/TCPP, prepared as described for phoPPF-3, was transferred to a sealed vial and subjected to stirring at 80 °C for 24 h. After the reaction, the product was collected by centrifugation, washed, and then dispersed in ethanol.

## Data availability

The data supporting the findings of this study are available within the Article and its Supplementary Information. Source data are provided with this paper.

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

## Acknowledgements

We acknowledge the financial support from the Natural Sciences and Engineering Research Council of Canada (NSERC), Fonds de recherche du Quebec-Nature et technologies (FRQNT), and National Natural Science Foundation of China (22475130). D.M. is also grateful to the Canada Research Chairs Program. S.K.M. thanks the Anusandhan National Research Foundation (ANRF), Government of India, for financial support under grant no. ANRF/IRG/2024/000078/ENS and ISIRD grant from IIT Ropar. We thank the Facility for Electron Microscopy Research (FEMR) of McGill University for help in electron microscopy operation and data collection. We thank Dr. Chuhan Fu in Ganjiang Innovation Academy for help in BET and XRD measurements. We also thank Prof. Aycan Yurtsever in Institut National de la Recherche Scientifique for his support on laser-integrated transmission electron microscopy.

## Author contributions

D.M., Y.L., and Y.W. conceived the concepts of photochemical synthesis of porphyrin MOFs and MOFs-based materials for photocatalysis. Y.L., Y.W., J.G., and R.Y. carried out the synthesis of samples and most of the characterizations. J.G. worked on the photocatalytic benzyl alcohol oxidation, UV-Vis, XRD, and DFT simulations for materials. K.K. had done the MD simulation for materials under the supervision of S.K.M. W.H. worked on the laser-integrated TEM analysis of materials. J.V. and W.H. carried out the 3D electron tomography reconstruction of phoPPF-3 under the supervision of A.M., D.M., and Y.L. Besides, Y.W., D.M., Y.L., Y.W., J.G., G.H., S.H., and Y.Z. all advised the experiments. Y.W., J.G., Y.L., and D.M. co-wrote the manuscript, and W.H. and K.K. also contributed to the manuscript.

## Competing interests

The authors declare no competing interests.
