## [Transparent Peer Review file · Nature Communications]

Room Temperature Photochemical Synthesis of Metal-Organic Frameworks for Enhanced Photocatalysis

Corresponding Author: Dr Dongling Ma

Version 0:

Reviewer comments:

Reviewer #1

(Remarks to the Author)

This manuscript reports a visible-light-driven, room-temperature ($\approx 15^\circ\text{C}$) strategy to synthesize MOFs—demonstrated in depth for a cobalt-porphyrin framework, phoPPF-3—that achieves high yield on short timescales, on/off photochemical control, and distinct coordination/morphology relative to solvothermal PPF-3. The authors further show enhanced thermal stability and improved photocatalysis (benzyl alcohol to benzaldehyde; H_2 evolution), and demonstrate generality to HKUST-1, ZIF-67, and ZIF-8. This is timely and impactful: it enables kinetic control and energy-efficient synthesis routes that conventional thermal methods struggle to access. A few comments for the authors to consider before acceptance in Nature Communications:

- Report kinetic parameters (e.g., pseudo-first-order k , initial rates) and TON/TOF for benzyl alcohol oxidation, and apparent quantum yields at defined wavelengths with calibrated photon flux (actinometry). Provide error bars and replicate statistics. Normalize rates by surface area and/or active-site metrics to enable fair comparison to literature and to PPF-3.
- The text describes CO_2 as oxidant; please substantiate this assignment. Include control atmospheres (N_2 , Ar, O_2 , CO_2), gas analysis (e.g., GC for CO, CO_2), and carbon balance. If CO_2 acts primarily as a proton/electron sink rather than a true oxidant, please clarify and adjust the scheme. Mechanistically, the discussion invokes PCET facilitated by the free-base porphyrin; consider ROS scavenger tests (e.g., tert-butanol, p-BQ, EDTA/TEOA), EPR spin-trapping, or transient absorption to strengthen the PCET/charge-transfer narrative.
- Provide photocatalysis controls (ligand only, metal salt only, physical mixture, dark, no BPY/PVP variants) and reusability data (multi-cycle H_2 tests with post-catalysis PXRD/XPS/DRS).
- The hourglass particles are central to the claim. Please quantify size/thickness distributions (SEMs are shown) alongside activity normalized per external surface area and, if feasible, facet- or region-selective activity probes (e.g., photodeposition of Pt to visualize charge-rich sites, Kelvin probe force microscopy, or spatially resolved photocurrent). A brief control on cubic morphologies (formed at $\geq 20^\circ\text{C}$) would help test morphology–activity correlations.
- Provide N_2 sorption (BET/DFT pore distribution) for phoPPF-3 vs PPF-3 to decouple electronic/coordination effects from possible surface-area/porosity differences that could drive rate changes. If porosity is low, discuss mass-transport contributions to observed kinetics.
- For phoHKUST-1, phoZIF-67, phoZIF-8, add yields vs time, light on/off traces, and side-by-side XRD fits (Rietveld if possible) to confirm lattice equivalence. Where practical, include a simple catalytic figure-of-merit (e.g., conversion or gas uptake) to show that photochemical synthesis does not compromise performance.
- The manuscript contrasts prior photothermal MOF syntheses with the present photochemical route. To avoid overstatement, please explicitly frame novelty as the first photon-initiated, room-temperature photochemical MOF synthesis that directs coordination and morphology without external photothermal agents, distinguishing it from light-induced heating approaches. A short paragraph reconciling this with cited prior art will strengthen the claim.
- Given the green-chemistry motivation, include a brief energy analysis (estimated kWh saved vs solvothermal), E-factor/PMI, and a note on scalability (e.g., gram-scale batch, continuous-flow photoreactor feasibility). The sunlight trial is compelling; a side-by-side energy/throughput table would underscore practical impact.

Reviewer #2

(Remarks to the Author)

The manuscript entitled "Room Temperature Photochemical Synthesis of Metal–Organic Frameworks for Enhanced Photocatalysis" presents a light-mediated approach for the synthesis of MOFs, which exhibit improved photocatalytic performance. Although photo-assisted routes for MOF synthesis are relatively uncommon, the mechanism of the synthetic process in this study has not been clearly demonstrated. Furthermore, the evidence to substantiate the enhanced photocatalytic activity of the photochemically synthesized MOFs is not sufficiently compelling. The data, in its current form, do not robustly support the central claims of the manuscript. Additionally, several material characteristics require more thorough evaluation. Therefore, I recommend major revision of this manuscript prior to consideration for publication.

Comments:

1. The mechanism underlying the light-controlled synthesis of MOFs remains unclear. Although the authors propose that the TCPP ligand acts as a photoactive component driving MOF formation under light, it should be noted that HKUST-1, ZIF-67, and ZIF-8 were synthesized without TCPP. Thus, it is unclear how the light-controlled process operates in these systems. Clarification is needed.
2. In Figure 3g, the authors present a proposed crystal structure for PPF-3 and phoPPF-3. However, the simulated PXRD pattern corresponding to these structures are not provided alongside the experimental data. Including the simulated pattern is essential to validate the structural model.
3. The authors attempt to demonstrate that Co^{2+} coordinates solely with carboxylate groups in phoPPF-3, but with both carboxylate and nitrogen-containing motif in PPF-3. The current evidence is insufficient to support this claim. More conclusive data, such as EXAFS, is needed to verify the coordination environment. Additionally, comparison of XPS, FT-IR, and UV-Vis spectra between free TCPP and Co^{2+} -TCPP would help confirm the chemical state of nitrogen in the MOFs.
4. The authors suggest that BPY serves as a linker connecting TCPP-Co coordination polymers to form the final MOF structure. To support this, ^1H NMR analysis of the digested MOF should be provided to determine the molar ratio between BPY and TCPP.
5. The interpretation of the TGA results appears misleading. The authors claim that the MOFs remain stable up to 415°C , yet the TGA curves indicate near-complete decomposition of organic ligands at this temperature. To substantiate the claimed thermal stability, PXRD patterns of the materials after heating to 415°C should be provided. Furthermore, temperature-dependent PXRD data are recommended to evaluate thermal stability comprehensively. Additional assessments of solvothermal and pH stability would also strengthen the study.
6. The authors assert that phoPPF-3 exhibits superior photocatalytic performance compared to PPF-3. However, the photocurrent response shows only a minimal difference. An explanation is warranted regarding how such a slight variation leads to a significant improvement in catalytic activity.
7. The enhanced photocatalytic performance of phoPPF-3 is attributed to the light-driven synthesis method. However, the altered coordination environment of Co^{2+} in phoPPF-3 compared to PPF-3 may itself account for the difference in activity, rather than the synthesis route.
8. In Figures 5o and 5p, the authors suggest that "electronic structure differences between TCPP and TCPP* influence coordination behavior with Co^{2+} ions, leading to structural divergence between phoPPF-3 and PPF-3". A more detailed explanation is needed regarding how these electronic changes specifically modulate the growth pattern and coordination geometry.
9. Essential details regarding the catalytic experiments, such as temperature, reaction scale, and substrate concentration, are missing from the figure captions. These should be included to ensure reproducibility and clarity.

Reviewer #3

(Remarks to the Author)

The manuscript reports a new synthesis technique for MOFs using light-driven strategy. This approach is demonstrated to be able to produce several types of MOFs which are more stable than the ones produced by conventional methods. The synthesized MOF also shows better photocatalytic activity compared to traditional MOFs. Therefore, I would recommend publication of the present work after the following points are addressed:

1. The authors claim that their synthesis method is a green approach (energy efficiency) compared to traditional methods. More evidence or demonstration is required, e.g., life cycle analysis.
2. The authors should provide the power density of the light source irradiated on the samples.
3. How is the photocatalytic stability of phoPPF-3 and conventional PPF-3?

Reviewer #4

(Remarks to the Author)

The authors report a visible light-driven method to synthesize many different MOFs within a short reaction time compared to the solvothermal method. The representative MOF, phoPPF-3, was chosen to study the mechanism of reaction in details. The authors also study the photocatalytic properties of phoPPF-3 in two types of reaction, oxidation and hydrogen production. In general, the reviewer finds this work interesting and indeed, it can be applied as a synthetic method to shorten

the reaction time in MOF synthesis. Therefore, the reviewer recommends publishing the work after a revision. Below are some concerns the authors may consider revising the manuscript.

1. Why did the authors choose 15°C for the photochemical synthesis? Did the authors investigate temperature dependence of the MOF synthesis? What about room temperature?
2. It may be helpful to show the structure of PPF-3 (and phoPPF-3). The authors mention that PPF-3 is a 2D structure. The reviewer does not understand why bpy was used. What is the role of bpy, pillar? If that is the case, which atom of bpy will bind to?
3. Minor correction: on page 8, line 199, should it be $\text{Co}_2(\text{COO})_4$ instead of $\text{Cu}_2(\text{COO})_4$?
4. Can the authors elaborate more on why "The phoPPF-3 with Co-free coordination in porphyrin center processed a weak charge repulsion between the MOF layers, which promotes H-stacking of phoPPF-3 units." Do the authors have evidence for this? The reviewer does not understand why no Co centers facilitate weak charge repulsion and thus promote H-stacking.
5. "These frameworks are further stabilized by BPY molecules in AB and AA stacking modes, ultimately yielding the final structures and morphologies of PPF-3 and phoPPF-3." The reviewer thinks it is unclear about the structure of the MOF (similar to question 2).
6. It is strange (but interesting) PPF-3 with Co centers binding to porphyrin units is inactive for hydrogen production compared to phoPPF-3. Did the authors try a control experiment by metalating phoPPF-3 with Co and then check its catalytic property?
7. The reviewer thinks it is important to have ICP analysis for PPF-3 and pho-PPF-3 to show different percentages of Co in the two MOFs.

Version 1:

Reviewer comments:

Reviewer #1

(Remarks to the Author)

The authors have satisfactorily addressed most of the comments raised by the reviewers through additional experiments, expanded mechanistic discussion, and improved clarity in the presentation of catalytic and structural data. In particular, the inclusion of additional control experiments, quantitative catalytic metrics (e.g., TON/TOF, error analysis), and improved discussion of reaction conditions and stability substantially strengthens the overall rigor of the study. Importantly, the revised manuscript now provides a more convincing justification of the proposed photochemical synthesis mechanism and better supports the claim that the light-driven synthetic route enables access to MOF structures and morphologies that are difficult to achieve under conventional solvothermal conditions. The demonstrated improvements in photocatalytic performance, combined with the broader applicability of the approach to other MOF systems, highlight the general impact of the methodology. Overall, this work presents a unique strategy for MOF synthesis under mild conditions with direct relevance to catalysis and materials design, and it merits publication.

Reviewer #2

(Remarks to the Author)

The authors have addressed most of the reviewers' concerns, and the revised manuscript is therefore suitable for publication.

Reviewer #3

(Remarks to the Author)

The authors have revised the manuscript and added the suggested data. Therefore, I would recommend publication of the present work.

Reviewer #4

(Remarks to the Author)

The reviewer read the revised versions of the manuscript and supporting information. The authors addressed comments from 4 reviewers and provided useful information, which significantly increases the manuscript's quality. Therefore, the reviewer recommends publishing this work.

Comments and responses

Reviewer #1:

This manuscript reports a visible-light-driven, room-temperature ($\approx 15\text{ }^{\circ}\text{C}$) strategy to synthesize MOFs—demonstrated in depth for a cobalt-porphyrin framework, phoPPF-3—that achieves high yield on short timescales, on/off photochemical control, and distinct coordination/morphology relative to solvothermal PPF-3. The authors further show enhanced thermal stability and improved photocatalysis (benzyl alcohol to benzaldehyde; H_2 evolution), and demonstrate generality to HKUST-1, ZIF-67, and ZIF-8. This is timely and impactful: it enables kinetic control and energy-efficient synthesis routes that conventional thermal methods struggle to access. A few comments for the authors to consider before acceptance in Nature Communications:

Response: Thank you for your positive comments on our manuscript. We feel that the questions raised by the reviewer below have significantly helped us to improve our scientific argument and thus the impact of our work on the development of MOF photocatalysis. Detailed responses to the questions are listed below.

Comment 1: Report kinetic parameters (e.g., pseudo-first-order k , initial rates) and ton/TOF for benzyl alcohol oxidation, and apparent quantum yields at defined wavelengths with calibrated photon flux (actinometry). Provide error bars and replicate statistics. Normalize rates by surface area and/or active-site metrics to enable fair comparison to literature and to PPF-3.

Response: Thank you for this critical and constructive comment. In response to the reviewer's comment, we have now enhanced the catalytic analysis in the revised manuscript by adding turnover number (TON)/turnover frequency (TOF) calculations, error bars with replication statistics, and surface-area-normalized rates.

To quantify the TON and TOF, we first determined the Co content in phoPPF-3 and PPF-3 using inductively coupled plasma optical emission spectroscopy (ICP-OES), which yielded values of 9.97wt% and 11.54wt%, respectively. The TON and TOF values were calculated to be 49.83 and 4.15 h^{-1} for PPF-3, and 76.70 and 6.39 h^{-1} for phoPPF-3, respectively (**Supplementary Table 4**). In addition, the apparent quantum yield (AQY), determined by calibrated photon flux (actinometry) at specific wavelengths, is a key metric for assessing process efficiency. The AQY of phoPPF-3 was also measured at 420 nm monochromatic light. Unfortunately, we admit that the AQY value in this study was relatively low, in line with many decent reports in the research areas of MOFs photocatalysis (Review: *Small Methods*, 2025, 9, 2401689). Furthermore, the detailed kinetic parameters (such as reaction order, initial rates) in studies concerning the photocatalytic oxidation of benzyl alcohol are often unreported (*J. Am. Chem. Soc.*, 2017, 139, 3513-3521; *ACS Catal.*, 2023, 21, 14346-14355; *Chem. Eng. J.*, 2024, 500, 156953), This is primarily because heterogeneous photocatalysis of benzyl alcohol is inherently more complex. Therefore, these kinetic parameters are generally difficult to establish effectively.

We have **added error bars and replicate statistics** for the benzyl alcohol oxidation data in the revised manuscript (**Fig. 7a** and **Supplementary Table 4**). The BET surface areas of phoPPF-3 and PPF-3 are 127.99 and $76.85\text{ m}^2\text{ g}^{-1}$, respectively (**Supplementary Table 5**). Therefore, the normalized rates by surface area of phoPPF-3 and PPF-3 are 0.87 and $0.96\text{ }\mu\text{mol h}^{-1}\text{ m}^{-2}$, respectively. In addition, the normalized rates by photocatalyst mass of phoPPF-3 and PPF-3 are 111.80 and $73.57\text{ }\mu\text{mol h}^{-1}\text{ g}^{-1}$. Moreover, we measured the Co content via ICP-OES for phoPPF-3 and PPF-3, which are 9.97 and 11.54 wt%, respectively. The normalized rates by Co content of phoPPF-3 and PPF-3 are 1121.36 and 637.52

Centre Energie Matériaux Telecommunications

1650, boulevard Lionel-Boulet
Varenes (Quebec) J3X 1S2 CANADA
T 514 228-6900

800, rue De La Gauchetiere Ouest, bureau 6900
Montreal (Quebec) H5A1K6 CANADA
T 514 228-7000

INRS.ca

$\mu\text{mol h}^{-1} \text{g}_{\text{Co}}^{-1}$. A comparative table has been created that includes our catalyst's performance normalized by photocatalyst mass alongside literature values normalized in the same way (**Table R1**). This allows for a direct and fair comparison of intrinsic activity.

Fig. 7a Benzaldehyde yield of PPF-3 and phoPPF-3 under the following photocatalytic conditions: 25 °C, 10 mg of photocatalyst, and the benzyl alcohol concentration of 4.82 mM.

Supplementary Table 4. Performance comparison of PPF-3 and phoPPF-3 in the photocatalytic benzyl alcohol oxidation reaction. The turnover number (TON) and turnover frequency (TOF) were calculated based on the Co content.

Sample	Benzaldehyde yield (%)			Mean value (%)	Standard deviation	TON	TOF (h^{-1})
	1	2	3				
PPF-3	9.18	11.25	10.01	~10.15	1.04	49.83	4.15
phoPPF-3	13.95	14.4	12.13	~13.49	1.20	76.70	6.39

Table R1. Comparison of benzyl alcohol oxidation rates of MOF-based photocatalysts.

Photocatalyst	Benzaldehyde production rate ($\mu\text{mol h}^{-1} \text{g}^{-1}$)	Reference
PPF-3	73.57	This work
phoPPF-3	111.80	This work

Zr-Co MOF@TBAPy	80.88	Appl. Catal., A , 2024, 682, 119826
UiO-66-NH ₂	177.92	Chem. Commun. , 2012, 48, 11656-11658
NH ₂ -MIL-125(Ti)	75.00	Appl. Catal., B , 2016, 187, 212-217
Ni-doped NH ₂ -MIL-125(Ti)	129.00	Appl. Catal., B , 2016, 187, 212-217
Ce-UiO-66	206.25	Sol. RRL , 2020, 4, 1900449

Comment 2: The text describes CO₂ as oxidant; please substantiate this assignment. Include control atmospheres (N₂, Ar, O₂, CO₂), gas analysis (e.g., GC for CO, CO₂), and carbon balance. If CO₂ acts primarily as a proton/electron sink rather than a true oxidant, please clarify and adjust the scheme. Mechanistically, the discussion invokes PCET facilitated by the free-base porphyrin; consider ROS scavenger tests (e.g., tert-butanol, p-BQ, EDTA/TEOA), EPR spin-trapping, or transient absorption to strengthen the PCET/charge-transfer narrative.

Response: Thank you for your insightful and constructive comment. In the initial Supporting Information, we mentioned that “*The mixture was degassed by N₂ and CO₂ for 20 min.*”. However, during the subsequent catalytic reaction, CO₂ was not continuously bubbled. To demonstrate the role of CO₂, we conducted a series of comparative experiments under different atmospheres (**Figure R1a**). Compared to reaction conducted under a CO₂ atmosphere, the slightly lower conversion observed under N₂ or Ar suggests that CO₂ likely functions as an oxidant that undergoes reduction itself (e.g., to CO). This inference is further supported by gas analysis, which detected trace amounts of CO (**Figure R1b**). Collectively, these results indicate that CO₂ indeed acts as an oxidant. Its introduction enhances the conversion rate of benzyl alcohol, primarily through a mechanism that promotes charge separation, thereby indirectly enhancing the oxidation ability. The decrease in conversion under an O₂ atmosphere confirms that O₂ negatively impacts the catalytic oxidation of benzyl alcohol, likely due to its strong oxidizing property leading to over-oxidation. In addition, carbon balance analysis revealed a significant molar discrepancy between the reduced product (CO) and the oxidized product (benzaldehyde). This indicates that CO₂ is not only a simple oxidant but also a multifunctional process aid, playing an important role in improving photocatalytic performance and regulating reaction pathways, such as inhibiting excessive oxidation, adjusting surface acidity and alkalinity. In addition, the presence of nitrate ions can also act as oxidants in the catalytic reaction, causing carbon imbalance. Accordingly, we revised the Supporting Information to provide a more precise description (**Page S5**), “*CO₂ acts as an oxidant to promote charge separation and as aids the process through regulating reaction pathways.*”

In our proposed mechanism, the free-base porphyrin core (H₂TCPP) plays a central role in facilitating a proton-coupled electron transfer (PCET) process, which is crucial for the efficient and selective oxidation of the substrate. The H₂TCPP acts as an integrated PCET platform. As a photosensitizer, it absorbs visible light to generate electron-hole pairs (e⁻/h⁺). Moreover, the two nitrogen atoms (N-H) at its center can participate in proton transfer, and the large π -conjugated system can stabilize electron and spin density, achieving synergistic management of protons and electrons, and therefore can be used as a PCET medium (*Chem. Commun.*, 2019, 55, 4925-4928; *J. Porphyrins Phthalocyanines*, 2021, 25,

674-682). The proposed PCET-driven catalytic cycle proceeds as follows: (1) **Photoexcitation**. H₂TCP* absorbs photons to form an excited state H₂TCPP*, which subsequently injects electrons into the metal-Co node, leaving oxidation state holes (h⁺) located on the porphyrin ring. (2) **Electron scavenging by CO₂**. The injected electrons are captured by adsorbed CO₂, driving its reduction (e.g., to

• CO₂⁻ or CO). This step quickly consumes electrons and suppresses e⁻/h⁺ recombination. (3) **PCET oxidation of benzyl alcohol**. The photogenerated h⁺ in the porphyrin ring attracts the hydroxyl oxygen atom of benzyl alcohol. Through a hydrogen-bonding network, the proton is transferred to a porphyrin nitrogen atom (N-H site), forming a transient protonated porphyrin. At the same time, the α-C-H bond electrons of benzyl alcohol are transferred to the porphyrin hole through the π-system, generating benzyl radicals. Proton and electron transfer occur in a single synergistic step, avoiding the generation of high-energy intermediates (such as benzyl alcohol radical cations) and significantly reducing energy barriers. The benzyl radical rapidly removes the second hydrogen (possibly through another PCET step), producing the final product benzaldehyde. (4) **Catalyst regeneration**. Protonated porphyrins release protons into oxidants, solvents or a basic species, restoring their free-base form.

To further strength the evidence for PCET mechanism of phoPPF-3 during photocatalysis, we evaluated the photocatalytic H₂ evolution using D₂O (**Figure R2**). The significantly lower reaction rate observed for phoPPF-3 in D₂O (919.0 μmol h⁻¹ g⁻¹) compared to that in H₂O (1684.6 μmol h⁻¹ g⁻¹) indicates that the proton transfer step is involved in the rate-determining step and is deeply coupled with the electron transfer (*Chem. Commun.*, 2009, 1757-1759), thus confirming the occurrence of a PCET mechanism.

Figure R1. (a) Benzaldehyde yield of phoPPF-3 under different atmospheres (CO₂, N₂, Ar, and O₂). (b) gas analysis of benzyl alcohol oxidation of phoPPF-3 under CO₂.

Figure R2. Photocatalytic H₂ evolution activities of phoPPF-3 tested in H₂O and D₂O over 12 h under the following photocatalytic conditions: 25 °C, 5 mg of photocatalyst, and a solution of 4.5 mL water with 0.5 mL triethanolamine.

Comment 3: Provide photocatalysis controls (ligand only, metal salt only, physical mixture, dark, no BPY/PVP variants) and reusability data (multi-cycle H₂ tests with post-catalysis PXRD/XPS/DRS).

Response: Thank you for your comment and suggestion. Incorporating these controls and reusability data will significantly strengthen our findings. During revision, we had performed several control experiments to validate the catalytic contribution of different components (**Supplementary Fig. 35**). Among the isolated precursors of Co(NO₃)₂, BPY, TCPP, and PVP, only TCPP exhibited hydrogen production activity (211.7 μmol g⁻¹ h⁻¹), attributed to its inherent photoactivity. The physical mixture of the components and PPF-3 showed significantly lower activities of 49.7 and 0.0 μmol g⁻¹ h⁻¹, respectively. These reductions may be due to the uncoordinated Co²⁺ and Co²⁺ coordinated at the porphyrin center acting as an electron scavenger, hindering efficient hydrogen evolution. In contrast, the phoPPF-3 (no BPY-PVP) and phoPPF-3 exhibited hydrogen production activities of 138.2 and 557.6 μmol g⁻¹ h⁻¹, respectively. These comparative results strongly suggest that the specific crystal structure of and the controlled incorporation of components within the phoPPF-3 are critical factors for its superior catalytic performance, facilitating efficient charge separation and utilization for hydrogen generation.

We further evaluated the reusability of phoPPF-3 for photocatalytic H₂ generation (**Supplementary Fig. 37a**). The phoPPF-3 exhibited a degree of reusability over short-term cycles. However, its long-term photocatalytic stability remains a significant challenge. Notably, the catalytic performance shows a marked decline by the fourth cycle. To investigate this degradation, we analyzed the UV-Vis spectra of phoPPF-3 before and after the photocatalytic reaction (**Supplementary Figure 37b**). For the post-reaction sample, the observed red shift in the Soret band and the merging of the Q bands indicate structural transformation. We attribute this to the potential incorporation of Co ions into the porphyrin ring, which is consistent with the known properties of PPF-3 and leads to diminished performance. Future work could explore strategies like microfluidic synthesis to mitigate this issue and enhance structural stability.

Supplementary Figure 35. Photocatalytic H₂ evolution activities of Co(NO₃)₂, BPY, TCPP, PVP, physical mixture (physical mixing of precursors for the synthesis of phoPPF-3), dark (phoPPF-3), phoPPF-3 (no BPY-PVP), PPF-3, and phoPPF-3 over 12 h under the following photocatalytic conditions: 25 °C, 10 mg of photocatalyst, and a solution of 18 mL water with 2 mL triethanolamine.

Supplementary Figure 37. (a) Cyclic stability testing of phoPPF-3 for photocatalytic H₂ production. The photocatalytic conditions for each cycle (0.5 h) were maintained at 25 °C with 10 mg of phoPPF-3 in a solution of 18 mL water and 2 mL triethanolamine. (b) UV-vis spectra of phoPPF-3 in the reaction suspensions before and after four cycles of photocatalytic H₂ production.

Comment 4: The hourglass particles are central to the claim. Please quantify size/thickness distributions (SEMs are shown) alongside activity normalized per external surface area and, if feasible, facet- or region-selective activity probes (e.g., photodeposition of Pt to visualize charge-rich sites, Kelvin probe force microscopy, or spatially resolved photocurrent). A brief control on cubic morphologies (formed at ≥ 20 °C) would help test morphology–activity correlations.

Response: Thank you for your comment and suggestion. The corresponding particle thickness distribution histogram of phoPPF-3 was provided based on the SEM image (**Supplementary Fig. 3 and Figure R3**). The average particle thickness, lateral dimension in the central region and end face edge length of the hourglass-like particles are ~ 352.0 , 354.1 and 478.7 nm, respectively, as determined by statistical analyses of the SEM image. Therefore, the external surface area of single phoPPF-3 particle is about $1.04 \text{ } \mu\text{m}^2$. Therefore, the normalize photocatalytic H_2 production activity by external surface area of phoPPF-3 is $\sim 536.2 \text{ } \mu\text{mol g}^{-1} \text{ h}^{-1} \mu\text{m}^{-2}$. However, due to the difficulty in measuring the specific number of particles per gram of phoPPF-3, we further normalized the activity using specific surface area. The BET surface area of phoPPF-3 is $127.99 \text{ m}^2 \text{ g}^{-1}$, and its corresponding normalize rate by surface area is $4.36 \text{ } \mu\text{mol h}^{-1} \text{ m}^{-2}$. In addition, the facet- or region-selective loading has been widely investigated in oxides or other materials (*Nat. Commun.*, 2013, 4, 1432; *Energy Environ. Sci.*, 2014, 7, 1369-1376; *Water Res.*, 2024, 262, 122101; *ACS Catal.*, 2025, 15, 11313-11325). We also attempted to load Pt via a photo-deposition method, however, no selective loading was observed.

We also synthesized the cubic phoPPF-3 at 25°C and evaluated its photocatalytic H_2 production. The cubic phoPPF-3 exhibited an activity of $520.2 \text{ } \mu\text{mol g}^{-1} \text{ h}^{-1}$, which is slight lower than that of hourglass-shaped phoPPF-3 ($557.6 \text{ } \mu\text{mol g}^{-1} \text{ h}^{-1}$). Therefore, the hourglass-shaped phoPPF-3 not only possessed a more uniform and regular morphology (compared to the sample synthesized at lower temperatures) but also exhibited a higher photocatalytic performance (compared to the sample synthesized at higher temperatures).

Supplementary Figure 3. (a) SEM image and (b) the corresponding particle thickness distribution histogram of phoPPF-3. The blue arrow in the inset indicates the thickness.

Figure R3. The lateral dimension in the central region (a) and end face edge length (b) distribution histograms of phoPPF-3 statistically obtained from Supplementary Figure 3a. The blue arrows in insets indicate the length measured.

Comment 5: Provide N₂ sorption (BET/DFT pore distribution) for phoPPF-3 vs PPF-3 to decouple electronic/coordination effects from possible surface-area/porosity differences that could drive rate changes. If porosity is low, discuss mass-transport contributions to observed kinetics.

Response: Thank you for your comment and suggestion. This is a crucial experiment to decouple the electronic/coordination effects from differences in surface area and porosity. We have conducted these measurements, and the results are presented in **Supplementary Fig. 38** and **Supplementary Table 5**. The BET surface areas of phoPPF-3 and PPF-3 are 127.99 m² g⁻¹ and 76.85 m² g⁻¹, respectively (Supplementary Fig. 38a). The average pore size of phoPPF-3 (10.29 nm) is larger than that of PPF-3 (6.71 nm). This difference is attributed to the appearance of a new pore size distribution peak at larger diameters (Supplementary Fig. 38b). The total pore volume of phoPPF-3 (0.33 m³ g⁻¹) is 1.54 times higher than that of PPF-3 (0.13 m³ g⁻¹).

For photocatalytic oxidation of benzyl alcohol, the observed differences in surface area, pore volume, and pore size distribution suggest that both intrinsic properties (electronic/coordination) and porosity/mass transport likely contribute together to the observed kinetic differences. The higher surface area and pore volume of phoPPF-3 provide more accessible sites for adsorption and potentially facilitate faster diffusion of molecules into the bulk of the material. This enhanced accessibility could contribute to the faster kinetics observed for phoPPF-3, independent of electronic effects. The presence of larger pores in phoPPF-3 suggests improved mass transport kinetics. This could be a significant factor in the observed rate enhancement. While the increased surface area and pore volume of phoPPF-3 likely play a substantial role in its faster kinetics, the electronic/coordination effects are also expected to enhance affinity and activation due to the PCET effect, which suggests a synergistic contribution. In particular, **for photocatalytic H₂ evolution**, the significantly higher activity of phoPPF-3 (557.6 μmol g⁻¹ h⁻¹) compared to PPF (which showed no activity) indicates that the electronic/coordination effects are the predominant driving force for its enhanced activity.

We have discussed these findings in the revised manuscript (**Page 18**), “*In addition, compared to PPF-3, the increased surface area (1.67-fold), pore size (1.53-fold), and pore volume (2.54-fold) of phoPPF-3 are likely to act synergistically with the intrinsically active sites, facilitating their more effective utilization and consequently enhancing photocatalytic performance (Supplementary Fig. 38 and Supplementary Table 5).*”.

Supplementary Figure 38. (a) Nitrogen adsorption–desorption isotherms and (b) the corresponding BJH pore size distributions of phoPPF-3 and PPF-3.

Supplementary Table 5. BET surface areas, average pore sizes, and total pore volumes of phoPPF-3 and PPF-3.

Sample	BET surface area (m ² g ⁻¹)	Average pore size (nm)	Total pore volume (m ³ g ⁻¹)
phoPPF-3	127.99	10.29	0.33
PPF-3	76.85	6.71	0.13

Comment 6: For phoHKUST-1, phoZIF-67, phoZIF-8, add yields vs time, light on/off traces, and side-by-side XRD fits (Rietveld if possible) to confirm lattice equivalence. Where practical, include a simple catalytic figure-of-merit (e.g., conversion or gas uptake) to show that photochemical synthesis does not compromise performance.

Response: Thank you for your excellent and critical suggestions. We appreciate the suggestion to assess the impact of photochemical synthesis on material performance. We have performed N₂ gas uptake measurements (at 0 to ~100 kPa and 77 K) for both the photochemically synthesized (phoHKUST-1, phoZIF-67, phoZIF-8) and conventionally synthesized materials (HKUST-1, ZIF-67, ZIF-8). As depicted in **Figure R4**, photoHKUST-1 exhibited a significantly lower N₂ uptake capacity (112.66 cm³ g⁻¹) compared to its conventionally synthesized counterpart (494.41 cm³ g⁻¹). This maybe due to the smaller crystal size of phoPPF-3 (**Fig. 6a**). In contrast, phoZIF-67 (581.61 cm³ g⁻¹) and phoZIF-8 (765.88 cm³ g⁻¹) showed slightly higher N₂ uptake capacities than ZIF-67 (489.02 cm³ g⁻¹) and ZIF-8 (687.39 cm³ g⁻¹), respectively, at ~100 kPa. These results indicate that while photochemical synthesis can yield materials with comparable or even improved gas adsorption properties for some MOFs, its effect is material-dependent, and not all photochemically synthesized materials maintain or exceed the performance of their conventionally synthesized analogues.

We performed systematic time-yield studies to quantify the kinetics of the photo-treatment process for phoHKUST-1 and phoZIF-67 (As ZIF-67 and ZIF-8 are structurally analogous). The resulting yield versus reaction time plots are presented in **Figure R5**. Both phoHKUST-1 and phoZIF-67 showed rapid generation within the first hour of irradiation, followed by a more gradual increase. While the conventional synthesized HKUST-1 and ZIF-67 showed a similar kinetic trend, their yields are consistently lower. This indicates that the photochemical synthesis method has a relatively small impact on the yield of these specific MOFs, likely attributable to the low photoexcitation response of their precursor ligands. Therefore, it's difficult to do the light on/off traces. We also performed XRD characterizations on the samples of phoHKUST-1, HKUST-1, phoZIF-67, and ZIF-67 collected after 1 h of reaction (**Figure R6**). For phoHKUST-1 and HKUST-1, their XRD patterns differ slightly from those of the final products (Fig. 6g), exhibiting more complex features that may be due to the transitional structures. However, the slight peak difference between phoHKUST-1 and HKUST-1 collected after 1 h of reaction may be due to the influence of light. In addition, for phoZIF-67 and ZIF-67, the XRD pattern of phoZIF-67 showed a narrower half peak width, suggesting a higher crystal growth rate

compared with that of ZIF-67. Therefore, these results further support that the photochemical synthesis can influence the crystallization pathway and kinetics of these MOFs.

Figure R4. Nitrogen adsorption isotherms of (a) HKUST-1 and phoHKUST-1, (b) ZIF-67 and phoZIF-67, and (c) ZIF-8 and phoZIF-8 at 77 K.

Figure R5. Time-dependent yield variations of (a) HKUST-1 and (b) ZIF-67 under light on and light off conditions.

Figure R6. XRD patterns of (a) phoHKUST-1 and HKUST-1, and (b) phoZIF-67 and ZIF-67 obtained after reaction for 1 h.

Comment 7: The manuscript contrasts prior photothermal MOF syntheses with the present photochemical route. To avoid overstatement, please explicitly frame novelty as the first photon-initiated, room-temperature photochemical MOF synthesis that directs coordination and morphology

Centre Énergie Matériaux Télécommunications

1650, boulevard Lionel-Boulet
Varenes (Québec) J3X 1S2 CANADA
T 514 228-6900

800, rue De La Gauchetière Ouest, bureau 6900
Montréal (Québec) H5A 1K6 CANADA
T 514 228-7000

without external photothermal agents, distinguishing it from light-induced heating approaches. A short paragraph reconciling this with cited prior art will strengthen the claim.

Response: Thank you for your suggestions. In the revised manuscript, we have added a dedicated paragraph in the Introduction section (**Page 3**), “*Herein, we report the first photon-initiated, room-temperature photochemical synthesis of MOFs that directs coordination and morphology without external photothermal agents, a key distinction from light-induced heating approaches. Specifically, we demonstrate*”. This highlights how our approach displayed specific photochemical mechanisms, rather than the photothermal effect, to precisely control the synthesis of MOFs.

Comment 8: Given the green-chemistry motivation, include a brief energy analysis (estimated kWh saved vs solvothermal), E-factor/PMI, and a note on scalability (e.g., gram-scale batch, continuous-flow photoreactor feasibility). The sunlight trial is compelling; a side-by-side energy/throughput table would underscore practical impact.

Response: Thank you for your excellent suggestion. We have conducted an analysis and added a new section titled “Techno-economic analysis of photochemical MOF synthesis.” in the revised manuscript (**Page 18 and 19**),

“Techno-economic analysis of photochemical MOF synthesis.

To evaluate industrial potential, we conducted a techno-economic analysis (TEA) comparing photochemical synthesis with conventional solvothermal route, focusing on energy, environmental metrics (E-factor and process mass intensity, PMI), and scalability. The photochemical route offers an estimated energy saving of approximately 92.4% - 94.3% over the solvothermal method, a benefit amplified if sunlight can be utilized. Conversely, laboratory-scale metrics (E-factor = 3675.48, PMI = 3676.48 for phoPPF-3, versus = 1837.24 and 1838.24 for PPF-3) currently indicate lower apparent economic efficiency due to high solvent usage inherent to milligram-scale batch synthesis. Such challenges are typical for early-stage synthetic development.

Scalability testing on a milligram-scale (~2 mg per batch) revealed that doubling the precursor concentration increased yield, but a threefold increase did not. This suggests that the light penetration became a limiting factor, and higher concentration may have hindered conversion due to increased light absorption by the reaction mixture, causing considerable light attenuation. But this can be solved by using a larger light source to increase the illumination spot falling on the reaction solution. Further considering the photochemical method operates under visible light and ambient conditions, which can be readily integrated into the continuous-flow photoreactor design, this method offers significant advantages for industrial scale-up. Thus, this TEA highlights the significant energy-saving benefits and practical potential of photochemical MOF synthesis (Supplementary Table 6), while also identifying key areas for optimization in waste reduction and reactor design to improve overall process sustainability and economy.”.

Energy Analysis:

We have estimated the energy consumption for our photochemical synthesis. The 420 nm LED illumination lamp has a power of 100 W and operates for 4 h per batch of phoPPF-3, resulting in an estimated energy consumption of 0.4 kWh per batch. For comparison, a typical solvothermal synthesis of PPF-3 requires heating at 80 °C for 24 h. Using a stirring hotplate with a rated power of 1.44 kW (HP88857100, Thermo Scientific), the heater's actual duty cycle under such moderate temperature conditions may be only 15% ~ 20%. The average hourly power consumption is therefore approximately

$1.44 \text{ kW} \times (0.15 \sim 0.20) = 0.22 \sim 0.29 \text{ kWh}$. Over 24 h, this translates to about 5.28 ~ 6.96 kWh per batch. Consequently, our photochemical route offers an estimated energy saving of approximately 92.4 – 94.3% per batch and 84.8 – 88.5% per gram compared to the conventional solvothermal method. Furthermore, if the synthesis of phoPPF-3 is conducted outdoors under sunlight, the energy saving would be even greater.

E-factor/PMI:

We have calculated the E-factor and PMI for photochemical synthesis. The E-factor is approximately 3675.48, and the PMI is about 3676.48. These metrics indicate low economy and high waste generation. The high E-factor and PMI in the laboratory-scale photochemical synthesis are primarily driven by the substantial quantities of solvents used for reaction, compounded by the inherent inefficiencies of small-scale laboratory operations. While these values are high, they reflect typical challenges in early-stage synthesis. We have detailed the calculation method in the Supporting Information (**Page S7**).

Scalability:

Our current synthesis has been demonstrated on a milligram-scale (~2 mg) batch basis. Doubling the precursor concentration led to an increase in yield. However, further increasing the precursor concentration threefold did not result in further significant improvement in yield. This due to the fact that the higher precursor concentration led to increased light absorption/scattering, causing strong light attenuation in the reaction solution and thereby hindering further conversion. The solution will be increasing the exposed reaction area under direct illumination by changing the light source to increase the illumination area, which accordingly requires the modification of the reactor. Although the current yield is low, we believe our photochemical approach is highly amenable to scalability. The use of visible light and room temperature conditions, coupled with the potential for continuous-flow photoreactor design, can offer significant advantages for industrial implementation. We have added a discussion on the feasibility of continuous-flow photoreactors in the revised manuscript.

Sunlight Trial and Practical Impact:

We agree that the sunlight trial is compelling. To underscore its practical impact and further highlight the energy efficiency, we have included a side-by-side table (**Supplementary Table 6**) comparing the energy consumption and throughput for our photochemical synthesis under different conditions (Xenon lamp vs. sunlight) against a representative solvothermal synthesis. This table clearly demonstrates the significant advantages in terms of energy savings and potential for practical application.

Supplementary Table 6. Energy consumption and throughput for the photochemical synthesis.

Method	Energy source	Reaction time (h)	Temperature (°C)	Energy consumption (kWh/mg product)	Product throughput (mg/batch)	Potential Throughput (mg/day)
Photochemical Synthesis	420 nm LED lamp	4	15	0.2	2.00	12.00 (6 batches)
Photochemical Synthesis	Sunlight	4	~30	0	0.73	1.46 (2 batches)

Solvothermal Synthesis	Hotplate	24	80	1.32~1.74	4.00	4.00 (1 batch)
----------	----	----	-----------	------	-------------------

Reviewer #2:

The manuscript entitled “Room Temperature Photochemical Synthesis of Metal–Organic Frameworks for Enhanced Photocatalysis” presents a light-mediated approach for the synthesis of MOFs, which exhibit improved photocatalytic performance. Although photo-assisted routes for MOF synthesis are relatively uncommon, the mechanism of the synthetic process in this study has not been clearly demonstrated. Furthermore, the evidence to substantiate the enhanced photocatalytic activity of the photochemically synthesized MOFs is not sufficiently compelling. The data, in its current form, do not robustly support the central claims of the manuscript. Additionally, several material characteristics require more thorough evaluation. Therefore, I recommend major revision of this manuscript prior to consideration for publication.

Response: Thank you for your insightful comments on our manuscript. We appreciate your suggestions for improving our scientific argument and data presentation. Detailed responses to the comments are listed below.

Comment 1: The mechanism underlying the light-controlled synthesis of MOFs remains unclear. Although the authors propose that the TCPP ligand acts as a photoactive component driving MOF formation under light, it should be noted that HKUST-1, ZIF-67, and ZIF-8 were synthesized without TCPP. Thus, it is unclear how the light-controlled process operates in these systems. Clarification is needed.

Response: Thank you for your insightful comment. The synthesis of HKUST-1, ZIF-67, and ZIF-8, which are well-established MOFs, typically does not involve light irradiation or the TCPP ligand. Our proposed mechanism for the light-controlled synthesis of phoPPF-3 specifically relies on the unique photoactivity of the TCPP ligand and its role as a photo-triggering component. In our system, photoexcitation of TCPP leads to significant electronic structure changes (as detailed in our discussion of dipole moment, polarizability, and electron density redistribution). These excited-state properties of TCPP, as we propose, alter its coordination behavior with Co^{2+} ions, favoring coordination to carboxylate groups rather than the porphyrin nitrogen. This photoinduced change in coordination preference is what dictates the different growth pathway and ultimately leads to the unique morphology of phoPPF-3. We also successfully synthesized phoMn-TCPP via photochemical method, further demonstrating the universality of our light-driven approach for MOF synthesis (**Supplementary Fig. 33**).

The synthesis of MOFs like HKUST-1, ZIF-67, and ZIF-8, in contrast, involves different precursors, ligands, and reaction mechanisms that do not rely on photoexcitation of a specific ligand to control coordination behavior or morphology. However, the reaction mixtures of these three MOFs become cloudy or even colored after mixing the precursors due to the rapid nucleation and growth of MOF crystallites (**Supplementary Fig. 32**). Consequently, light may be absorbed or scattered by these suspended seed crystals, thereby affecting their further growth. We revised the manuscript to more explicitly emphasize this system-specific nature of our light-controlled process (**Page 16 and 17**), “*The synthesis of these MOFs involves different precursors, ligands, and reaction mechanisms that do not*

rely on photoexcitation of a specific ligand to control coordination behavior or morphology. Light may be absorbed or scattered by MOF seed crystals, thereby affecting their further growth (Supplementary Fig. 32). Furthermore, we also synthesized *phoMn*-TCPP by substituting Mn^{2+} for Co^{2+} in the photochemical synthesis of *phoPPF*-3. For *phoMn*-TCPP, the UV-Vis spectral changes and the time-dependent yield variation under light on and light off conditions provided direct evidence for a photosynthesis process driven by TCPP photoexcitation (Supplementary Fig. 33).”

Supplementary Fig. 32. Photographs of the reaction mixtures for (a) HKUST-1, (b) ZIF-67, and (c) ZIF-8.

Supplementary Figure 33. (a) UV-vis spectra of TCPP, Mn-TCPP, and *phoMn*-TCPP. (b) Time-dependent yield variation of *phoMn*-TCPP under light on and light off conditions. Test conditions: the reaction substrate was three times the standard amount, and 2 mL of reaction solution was taken every 1 h for yield determination. Meanwhile, the light source was cycled between 1 h of light on and 1 h of light off. (c) XRD patterns of *phoMn*-TCPP and Mn-TCPP.

Comment 2: In Figure 3g, the authors present a proposed crystal structure for PPF-3 and *phoPPF*-3. However, the simulated PXRD pattern corresponding to these structures are not provided alongside the experimental data. Including the simulated pattern is essential to validate the structural model.

Response: Thank you for your suggestion. We would like to clarify that simulated PXRD patterns corresponding to our proposed crystal structures for PPF-3 and *phoPPF*-3 were indeed included in the initial submission. These are presented in **Supplementary Figs. 6 and 7**, and directly alongside the experimental PXRD data for comparison. We believe these patterns provide essential validation for our structural models. We are happy to provide them again for the reviewer’s convenience.

Supplementary Figure 6. Experimental and simulated XRD patterns of phoPPF-3, with a difference plot.

Supplementary Figure 7. Experimental and simulated XRD patterns of PPF-3, with a difference plot.

Comment 3: The authors attempt to demonstrate that Co^{2+} coordinates solely with carboxylate groups in phoPPF-3, but with both carboxylate and nitrogen-containing motif in PPF-3. The current evidence is insufficient to support this claim. More conclusive data, such as EXAFS, is needed to verify the coordination environment. Additionally, comparison of XPS, FT-IR, and UV-Vis spectra between free TCPP and Co^{2+} -TCPP would help confirm the chemical state of nitrogen in the MOFs.

Response: Thank you for your comment and suggestion. In our work, the UV-Vis spectra of these materials were employed to study the central coordination of porphyrin linker, which can provide strong evidence to support our proposed coordination modes (*J. Am. Chem. Soc.*, 2020, 142, 10331-10336; *Adv. Mater.*, 2015, 27, 7372-7378; *Adv. Funct. Mater.*, 2024, 34, 2315667; *J. Mater. Chem. B*, 2015, 3, 9340-9348; *J. Mater. Chem. A*, 2016, 4, 3933-3946). Compared to the free-base H_2TCPP , the Soret band

Centre Énergie Matériaux Télécommunications

1650, boulevard Lionel-Boulet
Varenes (Québec) J3X 1S2 CANADA
T 514 228-6900

800, rue De La Gauchetière Ouest, bureau 6900
Montréal (Québec) H5A 1K6 CANADA
T 514 228-7000

INRS.CA

of Co-TCPP exhibited a clear red shift and broadening. More importantly, the typical four Q-bands of H₂TCPP collapsed into one or two intensified bands, which is a characteristic signature of metalloporphyrin formation.

We acknowledge that advanced techniques like EXAFS can precisely determine the coordination geometry and the types of atoms directly bonded to the metal center. While EXAFS can provide valuable information about the first coordination sphere in general, it doesn't directly differentiate nitrogen and oxygen. If the material is a mixture (e.g., Co coordinating to both N and O in the same structure), the analysis becomes more complex. In our previous study on Co-TCPP MOF (Co-MOF-3), the EXAFS results show that "Co-MOF-3 has a similar peak position with porphyrin Co coordination center (containing CoN₄) and CoO at ca. 1.44 Å, which is mainly ascribed to the Co–N/O coordination at the first shell" (*Nat. Commun.*, 2021, 12, 1231). Therefore, we employed UV-Vis, FTIR, and XPS to verify the proposed coordination structure.

To confirm the chemical state of nitrogen in the MOFs, we performed **XPS, FTIR, and UV-Vis** analyses of H₂TCPP. As shown in **Supplementary Fig. 9**, the XPS N 1s spectrum of TCPP exhibits two types of nitrogen: a lower binding energy peak for the unprotonated nitrogen (–C=N–) and a higher energy peak of the protonated nitrogen (–NH–) (*Anal. Sci.*, 2005, 21, 635-639; *J. Porphyrins Phthalocyanines*, 2012, 16, 1235-1243). Critically, the retention of the unprotonated N (–C=N–) signal and the disappearance of the N-Co signal in the N 1s spectrum of phoPPF-3 as compared to TCPP and PPF-3 suggest the lack of coordination between pyrrolic N and Co ions in phoPPF-3, implying that Co doesn't bind at the porphyrin center. This finding is corroborated by the FTIR spectrum, where the N–H bands are retained and the N–Co features disappear in phoPPF-3 (**Supplementary Fig. 10**). Moreover, the UV-Vis spectra provide additional evidence (**Supplementary Fig. 12**). The spectrum of phoPPF-3 shows four absorption bands identical to those of TCPP, unlike PPF-3. This indicates that Co ions are not coordinated at the porphyrin center within phoPPF-3. Therefore, these results from XPS, FTIR, and UV-Vis spectroscopy confirm that Co²⁺ coordinates solely with the carboxylate groups in phoPPF-3, and not with the porphyrin nitrogen centers. Moreover, the **UV-Vis and FTIR spectra of H₂TCPP and Co²⁺-TCPP** were widely reported in the literature (*J. Mater. Chem. A*, 2016, 4, 3933-3946). Our results are highly consistent with these reported results (**Figures R7 and 8**), which further supports our proposed structures.

Supplementary Figure 9. XPS spectra of N 1s of phoPPF-3, PPF-3, and TCPP.

Centre Énergie Matériaux Télécommunications

1650, boulevard Lionel-Boulet
Varenes (Québec) J3X 1S2 CANADA
T 514 228-6900

800, rue De La Gauchetière Ouest, bureau 6900
Montréal (Québec) H5A 1K6 CANADA
T 514 228-7000

Supplementary Figure 10. FTIR spectra of phoPPF-3, PPF-3, and TCPP.

Supplementary Figure 12. UV-vis spectra of phoPPF-3, PPF-3, and TCPP in DMF/ethanol (3:1, v/v) solution. TCPP exhibited characteristic Q bands at 515, 550, 591, and 646 nm. PPF-3 showed a merged Q band at 548 nm, consistent with its UV-vis DRS result, suggesting the retention of Co-porphyrin coordination during dispersed solution. However, phoPPF-3 in solution showed Q bands at 517, 548, 591, 645 nm, suggesting a lack of central coordination of the TCPP linkers.

Figure R7. FTIR spectra (KBr) of (a) meso-tetra (4-methoxyphenyl) porphyrin (H₂TCPP) and (b) Cobalt(II) meso-tetra (4-carboxyphenyl) porphyrin (Co-TCPP) (*J. Mater. Chem. A*, 2016, 4, 3933-3946).

Figure R8. (a) UV-Vis spectrum (CH₃Cl) of meso-tetra (4-methoxyphenyl) porphyrin (H₂TCPP) and (b) UV-Vis spectra (THF) of Cobalt(II) meso-tetra (4-carboxyphenyl) porphyrin (Co-TCPP) (*J. Mater. Chem. A*, 2016, 4, 3933-3946).

Comment 4: The authors suggest that BPY serves as a linker connecting TCPP-Co coordination polymers to form the final MOF structure. To support this, ¹H NMR analysis of the digested MOF should be provided to determine the molar ratio between BPY and TCPP.

Response: Thank you for your valuable suggestion. To definitively support our proposed connectivity model, we have performed a ¹H NMR analysis on the fully digested solutions of both phoPPF-3 and PPF-3 (**Supplementary Fig. 13**). The integration of the relevant peak areas allowed us to determine the precise linker stoichiometry. The TCPP to BPY molar ratios were found to be 0.41 for PPF-3 and 0.58

for phoPPF-3 (Supplementary Fig. 13b and e). This direct stoichiometric evidence robustly validates our proposed connectivity scheme. Moreover, the presence of a small peak in Supplementary Fig. 13c, which signifies H within the porphyrin ring post acid dissociation, demonstrates the absence of intraring Co ion coordination in phoPPF-3. We have updated the manuscript to include these crucial quantitative findings (**Page 10**).

Supplementary Figure 13. The ^1H NMR spectra of the digested solutions of (a, b, and c) phoPPF-3 and (d, e, and f) PPF-3. (b and c) and (e and f) show magnified views of the areas indicated in (a) and (d), respectively.

Comment 5: The interpretation of the TGA results appears misleading. The authors claim that the MOFs remain stable up to 415°C , yet the TGA curves indicate near-complete decomposition of organic ligands at this temperature. To substantiate the claimed thermal stability, PXRD patterns of the materials after heating to 415°C should be provided. Furthermore, temperature-dependent PXRD data are recommended to evaluate thermal stability comprehensively. Additional assessments of solvothermal and pH stability would also strengthen the study.

Response: Thank you for your insightful comment and valuable suggestions. Upon re-examination of our characterization results, we acknowledge that our original statement regarding the thermal stability of phoPPF-3 up to 415°C was misleading. Our intention was to compare its stability with PPF-3 at temperatures below this temperature, rather than implying complete preservation of mass of organic components. We recognize that this critical distinction was not clearly conveyed, and the TGA, DTG, and TG/DTA-MS data indeed indicate partial ligand decomposition and potential structural compromise at this temperature. We have revised the relevant sections of the text to provide a more accurate reflection of the observed thermal behavior (**Page 11**), *“These results demonstrate the higher thermal stability of phoPPF-3 compared to PPF-3 below 415°C , highlighting its promise for long-term applications in a reasonably high temperature operation window”*. Furthermore, in response to the suggestion, we have performed PXRD analysis on both phoPPF-3 and PPF-3 after heating them to 415°C under the same conditions as the TGA. The new PXRD patterns, clearly show partial

Centre Energie Matériaux Télécommunications

1650, boulevard Lionel-Boulet
Varenes (Québec) J3X 1S2 CANADA
T 514 228-6900

800, rue De La Gauchetière Ouest, bureau 6900
Montréal (Québec) H5A 1K6 CANADA
T 514 228-7000

INRS.CA

decomposition of both phoPPF-3 and PPF-3, which is in good agreement with the TGA DTG, and TG/DTA-MS results (**Figure R9**). We fully agree that in-situ variable-temperature PXRD would be the most direct method to assessing thermal stability. Given the current limitations in instrument access, we have relied on a combination of TGA, DTG, TG/DTA-MS, and ex-situ PXRD of heat-treated samples to evaluate the thermal stability. We believe these analyses substantially address the concern. However, we plan to run the suggested temperature-dependent PXRD experiment in our future work to gain deeper insights.

In addition, we evaluated the solvothermal and pH stability of phoPPF-3 and PPF-3 via UV-Vis spectroscopy (**Figures R10 and 11**). Both phoPPF-3 and PPF-3 exhibited high solvothermal stability across 30-130 °C, showing no obvious spectral changes. However, phoPPF-3 showed lower pH stability than PPF-3, as evidenced by spectral shifts and/or intensity reductions. Under strong acidic conditions, protons may readily displace the metal centers in phoPPF-3, causing structural collapse. Similarly, strong bases may readily react with the metal ions in phoPPF-3 to form insoluble hydroxides, which also disrupt the phoPPF-3 framework.

Figure R9. XRD patterns of phoPPF-3 and PPF-3 after calcination at 415 °C under argon.

Figure R10. The solvothermal stability of (a) phoPPF-3 and (b) PPF-3 was evaluated by recording the UV-Vis absorption spectra of sample dispersions (5 mg in 5 mL DMF) at different temperatures.

Figure R11. The pH stability of (a and b) PPF-3 and (c and d) phoPPF-3 was evaluated by recording the UV-Vis absorption spectra of samples treated with aqueous solutions of different pH values. Specifically, a total of 5 mg of PPF-3 or PhoPPF-3 was added to a 10 mL glass bottle, followed by the addition of 5 mL of HCl or NaOH aqueous solutions with different pH. The mixture was stirred at room temperature for 2 h. After centrifugation, the solid sample was dissolved in DMF via ultrasonication. The pH stability of samples was subsequently characterized by UV-Vis spectroscopy.

Comment 6: The authors assert that phoPPF-3 exhibits superior photocatalytic performance compared to PPF-3. However, the photocurrent response shows only a minimal difference. An explanation is warranted regarding how such a slight variation leads to a significant improvement in catalytic activity.

Response: Thank you for your insightful comment. We agree that the photocurrent response for phoPPF-3 shows only a minimal difference compared to PPF-3. However, this subtle enhancement is significantly amplified in its catalytic performance in solution for the following major reasons. (1) Photocatalysis is a complex multi-step process occurring on suspended particles, involving not only charge generation and separation but also surface reaction kinetics and mass transfer. In contrast, photocurrent measurement is a simpler, more direct technique that primarily reflects the efficiency of charge separation and transport to the external electrode. (2) The photocurrent test is run in a photoelectrocatalysis system, where the measured photocurrent strongly depends on factors such as the catalyst-electrode contact resistance and MOF-MOF contact properties, in addition to catalytic reactions. Thus, photocurrent cannot be directly equated with the efficiency of photocatalysis in suspension-based systems, primarily because the reaction environment differs fundamentally from that of a thin-film electrode. In suspension, the photocatalyst presents a much larger accessible surface area, and reactants have significantly greater mass transport to the active sites of the MOF particles. Moreover, there may be some inconsistencies in many literature as well (*ACS Catal.*, 2018, 8, 2313-2325; *J. Mater. Chem. A*, 2019, 7, 25142-25154; *J. Phys. Chem. C*, 2012, 116, 5082-5089). (3) We propose that the key to the

superior photocatalytic performance of phoPPF-3 originates from changes at the interface, such as enhanced reaction kinetics or reactant adsorption.

Comment 7: The enhanced photocatalytic performance of phoPPF-3 is attributed to the light-driven synthesis method. However, the altered coordination environment of Co^{2+} in phoPPF-3 compared to PPF-3 may itself account for the difference in activity, rather than the synthesis route.

Response: Thank you for your insightful comment. The coordination environment of Co^{2+} in phoPPF-3 is indeed altered compared to PPF-3, and this difference in coordination undoubtedly plays a role in their catalytic activity. However, the light-driven synthesis method is the primary factor responsible for creating this specific and beneficial coordination environment in phoPPF-3, which then leads to superior performance. Conventional synthesis methods for similar MOFs typically lead to the metallation of the porphyrin core. Our light-driven approach, however, resulted in a distinct coordination state where Co^{2+} ions are primarily coordinated to the carboxylate groups, leaving the porphyrin cavity unoccupied. This specific outcome is a direct consequence of the photochemical activation and employed mild reaction conditions, rather than an inherent coordination preference of the precursor materials. While the altered coordination is the direct cause of the improved catalytic properties, it is the light-driven synthesis method that is responsible for establishing this specific, beneficial coordination environment, which conventional methods can not readily achieve. Furthermore, compared to PPF-3, the increased surface area (1.67-fold), pore size (1.53-fold), and pore volume (2.54-fold) of phoPPF-3 are likely to promote mass transfer, which in turn synergistically enhances photocatalytic performance. Therefore, the light-driven synthesis method is critical in yielding the superior photocatalytic performance of phoPPF-3.

Comment 8: In Figures 5o and 5p, the authors suggest that “electronic structure differences between TCPP and TCPP* influence coordination behavior with Co^{2+} ions, leading to structural divergence between phoPPF-3 and PPF-3”. A more detailed explanation is needed regarding how these electronic changes specifically modulate the growth pattern and coordination geometry.

Response: Thank you for raising this insightful point and requesting a more detailed mechanistic explanation. Our DFT calculations reveal that photoexcitation of TCPP to its excited state (TCPP*) induces a significant redistribution of electron density within the porphyrin ring, increasing its polarizability. Accordingly, this reduces the electron density and accessibility at the porphyrin nitrogen atoms. Consequently, during MOF assembly, Co^{2+} ions exhibit a lower thermodynamic driving force to coordinate with the porphyrin core in TCPP. Instead, coordination is preferentially directed toward the carboxylate groups. This fundamental shift in the primary coordination site—from porphyrin N in PPF-3 to carboxylate O in phoPPF-3—establishes a divergent initial coordination geometry, which is the origin of the structural divergence.

This altered coordination geometry directly dictates the subsequent crystal growth kinetics and morphology. Research indicates that orbital coupling in photoexcited layered structures can influence anisotropic growth rates (*Adv. Funct. Mater.*, 2017, 27, 1700925). In PPF-3, the metalloporphyrin unit promotes planar, sheet-like extension. In contrast, for phoPPF-3, the carboxylate-dominated coordination mode, under our specific light-driven synthesis conditions, guides the spatial orientation and linkage of TCPP* and BPY units differently. This promotes growth along multiple dimensions, ultimately leading to the observed three-dimensional, hourglass-shaped morphology. In summary, the electronic modification alters the primary metal-ligand binding site, which in turn modifies the coordination geometry and directs the assembly kinetics along a distinct pathway, resulting in the macroscopic morphological divergence.

We have rephrased the description in the revised manuscript (**Page 14**), “Specifically, this photoinduced electronic modification in TCPP* reduces the electron density or alters the accessibility of the porphyrin nitrogen atoms, making them less favorable for Co^{2+} coordination during the MOF formation. In contrast, in PPF-3, Co^{2+} preferentially coordinates to the porphyrin nitrogens of TCPP, leading to the formation of metalloporphyrin units. Furthermore, orbital coupling of layered structures in their electron-excited states can influence anisotropic growth rates, yielding unique layered morphologies.³³ This difference in preferred coordination geometry and binding sites directly determined the resulting MOF growth pattern.”.

Comment 9: Essential details regarding the catalytic experiments, such as temperature, reaction scale, and substrate concentration, are missing from the figure captions. These should be included to ensure reproducibility and clarity.

Response: Thank you for your valuable suggestion. We have revised all relevant figure captions to include the missing information, such as reaction temperature, reaction scale, and substrate concentration. For example, Fig. 7a now specifies (**Page 17**) “(a) Benzaldehyde yield of PPF-3 and phoPPF-3 under the following photocatalytic conditions: 25 °C, 10 mg of photocatalyst, and the benzyl alcohol concentration of 4.82 mM.” and “Supplementary Figure 34. Photocatalytic H_2 evolution activity of PPF-3 and phoPPF-3 over 12 h under the following photocatalytic conditions: 25 °C, 10 mg of photocatalyst, and a solution of 18 mL water with 2 mL triethanolamine.”. All figure captions have been updated accordingly.

Reviewer #3:

The manuscript reports a new synthesis technique for MOFs using light-driven strategy. This approach is demonstrated to be able to produce several types of MOFs which are more stable than the ones produced by conventional methods. The synthesized MOF also shows better photocatalytic activity compared to traditional MOFs. Therefore, I would recommend publication of the present work after the following points are addressed:

Response: Thank you for your positive comments on our manuscript. We appreciate your suggestions for improving the manuscript. According to your constructive suggestions, we have carefully revised our manuscript. Please find our detailed responses to the comments below.

Comment 1: The authors claim that their synthesis method is a green approach (energy efficiency) compared to traditional methods. More evidence or demonstration is required, e.g., life cycle analysis.

Response: Thank you for your insightful suggestions. We have conducted an energy analysis and added a new section titled “Techno-economic analysis of photochemical MOF synthesis.” in the revised manuscript.

Energy Analysis:

We have estimated the energy consumption for our photochemical synthesis. The 420 nm LED illumination lamp has a power of 100 W and operates for 4 h per batch of phoPPF-3, resulting in an

Centre Energie Matériaux Telecommunications

1650, boulevard Lionel-Boulet
Varenes (Quebec) J3X 152 CANADA
T 514 228-6900

800, rue De La Gauchetiere Ouest, bureau 6900
Montreal (Quebec) H5A1K6 CANADA
T 514 228-7000

INRS.CA

estimated energy consumption of 0.4 kWh per batch. For comparison, a typical solvothermal synthesis of PPF-3 requires heating at 80 °C for 24 h. Using a stirring hotplate with a rated power of 1.44 kW (HP88857100, Thermo Scientific), the heater's actual duty cycle under such moderate temperature conditions may be only 15% ~ 20%. The average hourly power consumption is therefore approximately $1.44 \text{ kW} \times (0.15 \sim 0.20) = 0.22 \sim 0.29 \text{ kWh}$. Over 24 h, this translates to about 5.28 ~ 6.96 kWh per batch. Consequently, our photochemical route offers an estimated energy saving of approximately 92.4 – 94.3% per batch and 84.8 – 88.5% per gram compared to the conventional solvothermal method. Furthermore, if the synthesis of phoPPF-3 is conducted outdoors under sunlight, the energy saving would be even greater. Therefore, the photochemical synthesis method is a green approach.

The following **Table R2** summarizes a comparative life cycle assessment for two synthetic methods of photochemical synthesis and conventional solvothermal synthesis. The comparison highlights the photochemical method's advantages in environmental friendliness, energy efficiency, and sustainability. Although photochemical synthesis is not yet mature, from a green chemistry and environmental impact perspective, photochemical synthesis imposes a significantly lower burden on the environment during the synthesis stage compared to conventional solvothermal methods.

Table R2. The comparison of the life cycle assessments of photochemical and solvothermal syntheses.

Stage	Key metrics	Solvothermal method	Photochemical method	Advantageous method
Raw material and energy input	Energy demand	Typically requires high temperatures (80 °C) maintained for 24 h, relying heavily on electrical energy.	Primarily driven by light sources (e.g., visible light), often operating at room temperature, drastically reducing energy input.	Photochemical
	Reaction time	Longer reaction times (hours to days), requiring sustained heating conditions.	Generally shorter reaction times, though rate is dependent on light intensity and absorption efficiency.	Photochemical
	Solvents and reagents	Often requires large volumes of organic solvents (e.g., DMF, EG, ethanol) as reaction media.	Often requires large volumes of organic solvents (e.g., DMF, EG, ethanol) as reaction media.	Similar
Waste and emissions	Waste Management	Generates significant volumes of liquid waste (containing organic solvents, unreacted salts, and byproducts).	Generates significant volumes of liquid waste (containing organic solvents, unreacted salts, and byproducts).	Similar
	GHG emissions	Indirectly produces higher CO ₂ emissions due to high-temperature heating (if electricity is fossil fuel-based).	Significantly lower CO ₂ footprint, especially if renewable energy is used for the light source.	Photochemical
Product separation and purification	Separation energy	Requires cooling, filtration, washing, and drying steps, often involving multiple washing cycles.	Separation process is simpler, potentially involving only filtration and washing (due to ambient reaction temperature).	Photochemical

Stage	Key metrics	Solvothermal method	Photochemical method	Advantageous method
	Product quality	Uniformity in crystal size and morphology is often easier to achieve through precise control of temperature and pressure.	Morphological control may rely on uniform light intensity/penetration, requiring careful tuning for consistency.	Solvothermal (Slightly better control)
Overall life cycle benefit	Environmental factor (E-factor)	Generally results in a lower E-Factor (1837.24).	Achieves a higher E-Factor (3675.48), due to laboratory-scale photochemical synthesis.	Solvothermal
	Scale-up challenges	Technology is mature and relatively easy to scale industrially (though energy consumption is a bottleneck).	Scale-up faces challenges related to light penetration depth, uniformity of illumination, and reactor design.	Solvothermal

Comment 2: The authors should provide the power density of the light source irradiated on the samples.

Response: Thank you for your suggestion. We have now included the power density of the light source irradiated on the samples in the revised manuscript (**Page 20**). The power density was $\sim 8 \text{ mW cm}^{-2}$ for the 420 nm LED light source.

Comment 3: How is the photocatalytic stability of phoPPF-3 and conventional PPF-3?

Response: Thank you for your insightful suggestion. We evaluated the cyclic stability of phoPPF-3 for photocatalytic H_2 generation (**Figure R12**). Notably, the conventional PPF-3 exhibited no activity in all four cycles for H_2 evolution. In contrast, phoPPF-3 exhibited significantly measurable photocatalytic activity over four cycles. However, its stability was poor, as evidenced by a significant performance drop after the second cycle. The reason may be that after photocatalytic reaction, a portion of Co^{2+} ions gradually became coordinated into the porphyrin center which is more thermodynamically stable, leading to the observed reduction in photocatalytic performance.

Figure R12. Cyclic stability testing of phoPPF-3 and PPF-3 for photocatalytic H₂ production. The photocatalytic conditions for each cycle were maintained at 25 °C with 10 mg of photocatalyst in a solution of 18 mL water and 2 mL triethanolamine.

Reviewer #4:

The authors report a visible light-driven method to synthesize many different MOFs within a short reaction time compared to the solvothermal method. The representative MOF, phoPPF-3, was chosen to study the mechanism of reaction in details. The authors also study the photocatalytic properties of phoPPF-3 in two types of reaction, oxidation and hydrogen production. In general, the reviewer finds this work interesting and indeed, it can be applied as a synthetic method to shorten the reaction time in MOF synthesis. Therefore, the reviewer recommends publishing the work after a revision. Below are some concerns the authors may consider revising the manuscript.

Response: Thank you for your positive comments on our manuscript. We appreciate your suggestions for improving the manuscript. We have carried out additional experiments and carefully revised our manuscript. Detailed responses to the questions are listed below.

Comment 1: Why did the authors choose 15°C for the photochemical synthesis? Did the authors investigate temperature dependence of the MOF synthesis? What about room temperature?

Response: Thank you for your comment. In our initial manuscript, we presented a study investigating the effect of reaction temperature on the morphology of phoPPF-3 (**Supplementary Fig. 20**). In the main text, we mentioned that “*The effect of reaction temperature on the morphology of phoPPF-3 was further investigated by synthesizing MOFs at different temperatures (0, 5, 10, 15, 20, and 25 °C) under 420 nm LED illumination and analyzed the products using SEM (Supplementary Fig. 20). At lower temperatures (0-5 °C), phoPPF-3 formed interconnected fibrous filaments. As the temperature increased to 10 °C, 3D MOF structures emerged alongside fiber-like assemblies. At 15 °C, the product exhibited a regular*

hourglass morphology, while temperature above 20 °C yielded cubic structures. These findings highlight the pronounced influence of reaction temperature on the morphology of phoPPF-3. The morphology variations can be attributed to temperature-dependent reactant diffusion rates, which modulate anisotropic growth kinetics and ultimately dictate the final morphology.” In addition, we also synthesized the cubic phoPPF-3 at 25 °C and evaluated its photocatalytic H₂ production. The cubic phoPPF-3 exhibited an activity of 520.2 μmol g⁻¹ h⁻¹, which is slight lower than that of hourglass-shaped phoPPF-3 at 15 °C (557.6 μmol g⁻¹ h⁻¹). Therefore, the hourglass-shaped phoPPF-3 not only possessed a more uniform and regular morphology but also exhibited a higher photocatalytic performance.

Supplementary Figure 20. SEM images of phoPPF-3 synthesized at temperatures of (a) 0 °C, (b) 5 °C, (c) 10 °C, (d) 15 °C, (e) 20 °C, and (f) 25 °C. The insets in (a)-(d) present zoomed-in SEM images of the selected areas.

Comment 2: It may be helpful to show the structure of PPF-3 (and phoPPF-3). The authors mention that PPF-3 is 2D structure. The reviewer does not understand why bpy was used. What is the role of bpy, pillar? If that is the case, which atom bpy will bind to?

Response: Thank you for raising this important point, which allows us to clarify the structural dimensionality. The primary network formed between Co²⁺ and TCPP is indeed two-dimensional, constituting a layered sheet. Crucially, the 4,4'-bipyridine (BPY) ligand functions as a pillaring agent. Each BPY molecule coordinates through its nitrogen atoms to the Co²⁺ ions in two adjacent layers, thereby connecting these 2D sheets in the third dimension to form the final three-dimensional framework of PPF-3, as established in the literature (e.g., *J. Am. Chem. Soc.*, 2016, 138, 6924-6927, **Figure R13**).

This core structural role of BPY as a pillar, binding directly to the Co²⁺ ions of adjacent layers, remains identical in phoPPF-3, resulting in the same underlying 3D coordination network. The observed **morphological difference**, where PPF-3 appears as 2D micro/nanosheets while phoPPF-3 forms 3D hourglass-shaped crystals, arises from distinct crystal growth kinetics during synthesis, not from a change in the fundamental pillared-layer connectivity.

PUBLISHED DATA FIGURE REDACTED

Figure R13. Crystal structure of PPF-3 nanosheets (*J. Am. Chem. Soc.*, 2016, 138, 6924–6927). (a) Structures of $\text{Co}_2(\text{COO})_4$ paddlewheel metal node, TCPP ligand, and the corresponding “checkerboard-like” 2D layered sheet. (b) The layered sheets are further pillared by BPY molecules in an AB stacking pattern to form the final structure. Reprinted with permission from American Chemical Society.

Comment 3: Minor correction: on page 8, line 199, should it be $\text{Co}_2(\text{COO})_4$ instead of $\text{Cu}_2(\text{COO})_4$?

Response: Thank you for pointing out this error. It should be $\text{Co}_2(\text{COO})_4$. We have corrected this typo in the revised manuscript.

Comment 4: Can the authors elaborate more on why "The phoPPF-3 with Co-free coordination in porphyrin center processed a weak charge repulsion between the MOF layers, which promotes H-stacking of phoPPF-3 units." Do the authors have evidence for this? The reviewer does not understand why no Co centers facilitates weak charge repulsion and thus promotes H-stacking.

Response: Thank you for your insightful comment. The occurrence of H- or J-aggregation can be identified by the characteristic changes in the Soret band (~400-450 nm) of the UV-Vis spectra. As shown in **Supplementary Fig. 12**, in PPF-3, a strong J-aggregation was observed at 433 nm, along with its relatively thinner, sheet-like morphology. However, in phoPPF-3, a broader Soret band with a weak J-aggregation was observed, along with its relatively thick morphology. The literature reported that “the central metal coordination of porphyrins, resulting in a charge repulsion between the MOF layers and reducing “face-to-face stacking” (“H-stacking”) between the layers” and “The metal-coordinated Cu-TCPP(BA) sheet had a strong charge repulsion between the layers, which hindered the H-stacking of sheets” (*J. Am. Chem. Soc.*, 2016, 138, 6924–6927). In our system, the absence of Co^{2+} coordination in the central porphyrin cavity of phoPPF-3 can reduce the electrostatic repulsion between the MOF layers. This weaker repulsion facilitates closer interlayer contacts and enhances face-to-face stacking. Therefore, we concluded that “The phoPPF-3 with Co-free coordination in porphyrin center processed a weak charge repulsion between the MOF layers, which promotes H-stacking of phoPPF-3 units.” We have revised the manuscript to clarify these points (**Page 10**), “Moreover, PPF-3 showed strong J-aggregation at 433 nm, along with its relatively thinner, sheet-like morphology. In contrast, phoPPF-3 exhibited a broader Soret band, indicative of weak J-aggregation, along with a thicker morphology. Therefore, the phoPPF-3 with Co-free coordination in porphyrin center processed a weak charge repulsion between the MOF layers, which promotes H-stacking of phoPPF-3 units.”.

Supplementary Figure 12. UV-vis spectra of phoPPF-3, PPF-3, and TCPP in DMF/ethanol (3:1, v/v) solution. TCPP exhibited characteristic Q bands at 515, 550, 591, and 646 nm. PPF-3 showed a merged Q band at 548 nm, consistent with its UV-vis DRS result, suggesting the retention of Co-porphyrin coordination during dispersed solution. However, phoPPF-3 in solution showed Q bands at 517, 548, 591, 645 nm, suggesting a lack of central coordination of the TCPP linkers.

Comment 5: "These frameworks are further stabilized by BPY molecules in AB and AA stacking modes, ultimately yielding the final structures and morphologies of PPF-3 and phoPPF-3." The reviewer thinks it is unclear about the structure of the MOF (similar to question 2).

Response: Thank you for your comment. The 2D sheet structure (formed by Co^{2+} coordination with TCPP) is the fundamental building block. In PPF-3, these 2D sheets are then arranged in the **AB stacking mode**, stabilized by BPY molecules acting as pillars (**Fig. 3g**). This AB stacking of the 2D sheets, further mediated by BPY, results in the overall 2D micro/nanosheet morphology of PPF-3. In addition, in the synthesis of PPF-3, the added surfactant molecule, polyvinylpyrrolidone (PVP), was selectively attached onto the surface of PPF-3, which controlled the vertical growth of PPF-3 crystals, resulting in the formation of 2D PPF-3 micro/nanosheets (*J. Am. Chem. Soc.*, 2016, 138, 6924-6927). In contrast, for phoPPF-3, the 2D sheets (formed by Co^{2+} coordination with TCPP) do not feature Co^{2+} coordination in porphyrin center. This absence, coupled with BPY acting as a pillar that coordinates to the Co^{2+} ions of the paddle-wheel nodes, promotes the **AA stacking mode** due to strong coordination force between functional groups and metal nodes.

We have revised the text to more explicitly explain the structures of PPF-3 and phoPPF-3 (**Page 10**). "TCPP ligands coordinate with Co metal nodes under varying conditions, forming slightly varied 2D "chessboard-like" sheets (Fig. 3g). These two different sheet structures are further pillared by BPY molecules in AB and AA stacking modes, respectively, ultimately yielding the final 3D network structures of PPF-3 and phoPPF-3. The Co-free coordination in porphyrin center, coupled with BPY acting as a pillar that coordinates to the Co^{2+} ions of the paddle-

wheel nodes, promotes an AA stacking pattern of phoPPH-3 due to strong coordination force between functional groups and metal nodes.”.

Fig. 3g Schematic of PPF-3 and phoPPF-3 molecular assembly, depicting the formation of layered unit structures and 3D architectures with AA and AB stacking modes.

Comment 6: It is strange (but interesting) PPF-3 with Co centers binding to porphyrin units is inactive for hydrogen production compared to phoPPF-3. Did the authors try a control experiment by metalating phoPPF-3 with Co and then check its catalytic property?

Response: Thank you for your insightful comment and suggestion. To address this, we have performed the suggested control experiment. We metalated phoPPF-3 with Co by heating a solution of phoPPF-3 and $\text{Co}(\text{NO}_3)_2$ at 80 °C for 12 h, yielding a sample we referred to as “phoPPF-3/ Co^{2+} ”. We then evaluated its catalytic performance for hydrogen production under the same reaction conditions as phoPPF-3 (**Supplementary Fig. 36**). Meanwhile, we also evaluated the catalytic performance of the phoPPF-3+ Co^{2+} sample, which was obtained by simple physical mixing of phoPPF-3 and $\text{Co}(\text{NO}_3)_2$ in solution.

The results show that phoPPF-3/ Co^{2+} exhibited negligible hydrogen production activity ($312.8 \text{ } \mu\text{mol g}^{-1} \text{ h}^{-1}$). This value is slightly higher than that of phoPPF-3+ Co^{2+} ($269.4 \text{ } \mu\text{mol g}^{-1} \text{ h}^{-1}$) but significantly lower than that of phoPPF-3 ($557.6 \text{ } \mu\text{mol g}^{-1} \text{ h}^{-1}$). These findings strongly suggest that the crystal structure of phoPPF-3 is indeed a critical factor for its superior catalytic performance, rather than simply the presence of Co centers within a porphyrin framework. Furthermore, the presence of Co^{2+} demonstrated a negative effect on the photocatalytic performance for hydrogen production, which is consistent with the poor photocatalytic performance observed for PPF-3 ($0.0 \text{ } \mu\text{mol g}^{-1} \text{ h}^{-1}$). The underlying reason may be that uncoordinated Co^{2+} and Co^{2+} coordinated at the porphyrin center act as an electron scavenger, capturing photogenerated electrons and impeding the hydrogen generation process. We also included the details of these control experiments in the revised Supporting Information (**Page S26**).

Supplementary Figure 36. Photocatalytic H₂ evolution activity of phoPPF-3, phoPPF-3/Co²⁺, and phoPPF-3+Co²⁺ under the following photocatalytic conditions: 25 °C, 10 mg of photocatalyst, and a solution of 18 mL water with 2 mL triethanolamine. The phoPPF-3/Co²⁺ sample was synthesized by heating a solution of phoPPF-3 and Co(NO₃)₂ at 80 °C for 12 h, while the phoPPF-3+Co²⁺ sample was obtained by simple physical mixing of the two components in solution.

Comment 7: The reviewer thinks it is important to have ICP analysis for PPF-3 and pho-PPF-3 to show different percentage of Co in the two MOFs.

Response: Thank you for your valuable suggestion. We have performed ICP-OES analysis for both PPF-3 and phoPPF-3. The results confirm a Co content of 11.54wt% and 9.97wt% in PPF-3 and phoPPF-3, respectively. These results closely match the theoretical values calculated for the proposed unit structures (12.36wt% for PPF-3 unit and 9.68wt% for phoPPF-3 unit). The slight discrepancy is likely attributed to variations in the PVP content between the two samples. Thus, this ICP-OES results are consistent with our proposed structures and will be included in the revised manuscript (**Page 10**), “*Additionally, the Co content in phoPPF-3 (9.97wt%) is slightly lower than that in PPF-3 (11.54wt%). These results closely match the theoretical values calculated for the proposed unit structures (12.36wt% for PPF-3 unit (Co:TCPP:BPY=3:1:3) and 9.68wt% for phoPPF-3 unit (Co:TCPP:BPY=2:1:2).*”.